DESY-25-095

# Orthogonality of Q-Functions up to Wrapping in Planar $\mathcal{N} = 4$ Super Yang–Mills Theory

Till Bargheer[1], Carlos Bercini[1], Andrea Cavaglià[2], Davide Lai[1], Paul Ryan[1]

[1] *Deutsches Elektronen-Synchrotron DESY, Notkestr. 85, 22607 Hamburg, Germany*
[2] *Dipartimento di Fisica, Università di Torino and INFN, Sezione di Torino, Via P. Giuria 1, 10125, Torino, Italy*

## Abstract

We construct orthogonality relations in the Separation of Variables framework for the $\mathfrak{sl}(2)$ sector of planar $\mathcal{N} = 4$ supersymmetric Yang–Mills theory. Specifically, we find simple universal measures that make Q-functions of operators with different spins vanish at all orders in perturbation theory, prior to wrapping corrections. To analyze this rank-one sector, we relax some of the assumptions thus far considered in the Separation of Variables framework. Our findings may serve as guidelines for extending this formalism to other sectors of the theory as well as other integrable models.

# 1 Introduction

The presence of integrability in planar $\mathcal{N} = 4$ SYM has led to the development of powerful non-perturbative tools for studying correlation functions. For two-point functions, the Quantum Spectral Curve [1, 2] allows to extract the conformal dimension of any single-trace local operator, at any value of the 't Hooft coupling, including strong coupling [3], from a handful of so-called Q-functions. For three- and higher-point correlation functions there is the Hexagon framework [4, 5], valid even beyond the planar limit [6, 7], and with some modifications also in $\mathcal{N} = 2$ contexts [8, 9].

The Hexagon framework is highly efficient for asymptotically large operators. For short operators, contributions from so-called mirror particles render its application much more difficult. In this sense, the Hexagon framework is analogous to the Asymptotic Bethe Ansatz for the spectral problem [10]. For the finite-size spectrum, the latter needs to be supplemented with Lüscher corrections via the Thermodynamic Bethe Ansatz [11–13], which then leads to the Quantum Spectral Curve, with all finite-size corrections elegantly taken into account. Additionally, features of conformal data such as analyticity in spin [14] are obscured in the Hexagon approach, while being completely manifest in the Quantum Spectral Curve [15–19].

It is thus highly desirable to find a formalism that allows one to compute correlation functions directly in terms of the Quantum Spectral Curve Q-functions. In conventional integrable spin chains, this is achieved by the Separation of Variables (SoV) method, which was pioneered by Sklyanin [20] and extensively developed in $\mathfrak{sl}(2)$-based models in [21, 22]. The formalism has undergone rapid advancement in recent years, in particular for higher-rank symmetries [23–30]. It allows one to cast the wave functions of the Hamiltonian eigenstates in a factorized product of single-variable building blocks, the Q-functions, from which correlation functions can then be computed.

Separation of Variables computations have already yielded impressive results for correlation functions in $\mathcal{N} = 4$ SYM [31–34] and related theories [35]. The current state-of-the-art at weak coupling is the SoV expression for three-point correlators involving two BPS operators and one non-BPS operator in the $\mathfrak{su}(2)$ and $\mathfrak{sl}(2)$ sectors [36].

A key building block for physical observables such as correlation functions [29] is a scalar product on the space of states realized in terms of Q-functions, i.e. a relation that becomes zero for distinct states. This relation takes the form of a determinant of inner products of Q-functions, where the inner products are integrals against simple measures, which depend on the representation. A modern method to construct the orthogonality relation is the Functional Separation of Variables formalism [26], which is based on the so-called Baxter TQ equation.

In principle, since the main ingredient needed for the Functional Separation of Variables formalism is just the TQ equation, one can apply it to arbitrarily high orders in perturbation theory in planar $\mathcal{N} = 4$ SYM, at least in the closed rank-one bosonic sectors.[1] However, unlike in rational spin chains, where the Q-functions are polynomial,[2] and for which most of the SoV development has been carried out, numerous complications arise. The Q-functions start receiving quantum corrections and are no longer polynomial. The Baxter TQ equations and the measures also receive corrections. Additionally, beyond leading order in perturbation theory, we do not have an operatorial description of integrability in $\mathcal{N} = 4$ SYM, meaning

---

[1]The Functional Separation of Variables formalism has only been fully developed for rational bosonic spin chains based on $\mathfrak{sl}(n)$ and $\mathfrak{su}(n)$ [28, 29] (see [37] for a review), and there are no closed bosonic subsectors of planar $\mathcal{N} = 4$ SYM at higher orders in perturbation theory besides the rank-one $\mathfrak{su}(2)$ and $\mathfrak{sl}(2)$ sectors.

[2]For non-compact highest-weight representations, not all Q-functions are polynomial, but all physical quantities can still be described by just the polynomial Q-functions, at least in the examples considered in the literature, see [28].

that operatorial techniques such as those of [23, 24, 38] are not available. More abstractly, the fact that finite-coupling transfer matrices and Q-operators [21, 39, 40] are not yet formulated is related to the unknown quantum algebra underlying the $\mathcal{N} = 4$ SYM model. A full understanding of the underlying quantum algebra should indicate an algebraic path towards the corresponding Q-operators.[3]

In view of these difficulties, in this paper, we take an explorative approach to construct the desired orthogonality relation. We focus on the $\mathfrak{sl}(2)$ sector, whose operators are described by their twist $L$ and spin $S$ and take the schematic form

$$\text{Tr}(D^S Z^L) + \text{permutations} \tag{1.1}$$

where $Z$ is a complex scalar and $D$ is a covariant derivative. Based on observations for twist-two and twist-three states at low loop orders, we propose an orthogonality relation for all $\mathfrak{sl}(2)$ states to all orders in perturbation theory before wrapping effects kick in. The orthogonality is formulated in terms of finite-coupling measures (3.12) together with a set of ever-growing matrices (3.5) of Q-functions whose determinant (3.7) vanishes when the states have different values of spin.

However, there is one important aspect in our SoV-type expressions that needs to be emphasized. For reasons we do not yet fully understand, our proposed orthogonality (3.7) breaks for states of equal spin. Since in $\mathfrak{sl}(2)$, the spin is a proxy for leading-order degeneracy between the states, another way we can state this fact is: *Our SoV expressions are valid only for operators that are not degenerate at leading order.* For twist-two operators, there are no such degeneracies, so this problem is absent. On the contrary, higher-twist operators do present this problem. For example, at twist three, there are three degenerate states with spin six at leading order. We observe that our formulas correctly reproduce orthogonality between these degenerate states and all the other twist-three operators (up to wrapping), but they give a non-vanishing overlap when contracting the degenerate states among themselves.

In the last section of this work, we present a possible solution to this puzzle, by considering generalizations to the usual assumptions considered in the SoV framework. This allows us to establish a new strategy to deduce orthogonality in terms of Q-functions that we believe could also be applicable in more general integrable systems, where the Separation of Variables approach has proven to be elusive.

In Section 2 we review the SoV approach for twist-two operators, recalling the results for two-point functions at leading and next-to-leading order. We show how the problems involving the appearance of new degrees of freedom naturally emerge at N²LO, and how we solve them.

In Section 3 we present the main results of this work, where we generalize the expressions of twist-two to higher-twist operators at any order in perturbation theory before wrapping corrections.

In Section 4 we generalize the usual assumptions of the SoV framework, and consider a more general type of Q-function for twist-two operators. This results in a new SoV expression at N²LO that we believe can resolve the puzzle of orthogonality among degenerate states and yield new SoV representations for integrable models.

Finally, in Section 5 we summarize all our results and comment on the several limits of strong coupling, large spin and large charge that could be accessible by our SoV proposal.

---

[3]There has been some recent progress towards the related quantum algebra of the spin chain excitation picture [41].

# 2 Orthogonality for Twist Two

## 2.1 Functional Separation of Variables at Leading Order

We will start by considering the simplest class of $\mathfrak{sl}(2)$ operators, the twist-two operators, which have the form $\mathrm{Tr}(ZD^SZ) + \mathtt{permutations}$. At leading order, the Q-functions that describe such operators are given by the solutions to the Baxter equation [42]

$$0 = \mathcal{B}_S \cdot Q_S \equiv \left(u + \frac{i}{2}\right)^2 Q_S(u+i) + \left(u - \frac{i}{2}\right)^2 Q_S(u-i) + (I_0(S) - 2u^2)Q(u)\,. \quad (2.1)$$

Here, we have introduced a finite-difference operator $\mathcal{B}_S$ called the *Baxter operator*, which acts on the Q-function by shifts of the spectral parameter. As we review below, $\mathcal{B}_S$ is a central object for the functional SoV method. The term that multiplies $Q(u)$ on the right-hand side is the *transfer matrix*, whose expansion coefficients are the state-dependent integrals of motion $I_j(S)$. In this example, $I_0(S) = 1/2 + S + S^2$ and $I_2(S) = -2$.

The solution space of the Baxter equation (2.1) is two-dimensional. A generic solution will be non-polynomial, but there is one unique polynomial solution

$$Q_S(u) = \prod_{n=1}^{S}(u - v_n)\,, \quad (2.2)$$

whose roots $v_n$ are the solution to the asymptotic Bethe equations recalled in Appendix A. We will come back to the other, non-polynomial solutions in Section 4.

Our goal is to find Separation of Variables type expressions for two-point functions in this $\mathfrak{sl}(2)$ sector of $\mathcal{N} = 4$ super Yang–Mills theory. Namely, we want to find simple universal measures that, when integrated against these Q-functions, result in the two-point correlation function of the corresponding operators. Since two-point functions of different states are zero, one simpler and often sought-after goal is to find measures that make Q-functions of different states *orthogonal* to each other. In the Functional Separation of Variables approach, the starting point for this orthogonality is the trivially vanishing expression

$$\int_{\mathbb{R}} \left[ Q_S(u) \overleftarrow{\mathcal{B}}_S Q_J(u) - Q_S(u) \overrightarrow{\mathcal{B}}_J Q_J(u) \right] \mu(u)du = 0, \quad (2.3)$$

where $\overleftarrow{\mathcal{B}}_S$ ($\overrightarrow{\mathcal{B}}_J$) act on the Q-function on their left (right), respectively, and $\mu(u)$ is an integration measure which will be fixed by imposing some essential properties. In fact, the trivial relation (2.3) can be turned into a useful relation if we can find an integration measure that is both state-independent and makes the Baxter operator self-adjoint, i.e. allows us to replace $\overleftarrow{\mathcal{B}}_S$ with $\overrightarrow{\mathcal{B}}_S$ under integration. This can be achieved by finding an $i$-periodic measure that allows us to freely shift contours in the imaginary direction, in order to move the shifts in $u$ from one Q-function to the other. For example,

$$\int_{\mathbb{R}} Q_S(u) \left(u + \frac{i}{2}\right)^2 Q_J(u+i)\mu(u)du \overset{u \to u-i}{=} \int_{\mathbb{R}} Q_S(u-i)\left(u - \frac{i}{2}\right)^2 Q_J(u)\mu(u)du\,. \quad (2.4)$$

This shifting of contours has been extensively considered in the SoV framework in many different contexts [26–28, 43], and its outcome is to recast equation (2.3) as

$$0 = \int_{\mathbb{R}} Q_S(u)(\overleftarrow{\mathcal{B}}_S - \overleftarrow{\mathcal{B}}_J)Q_J(u)\mu(u)du + \mathtt{res}_\mu = \Delta I_0 \int_{\mathbb{R}} Q_S(u)Q_J(u)\mu(u)du + \mathtt{res}_\mu\,, \quad (2.5)$$

where we used the explicit expression of the leading-order Baxter operator (2.1) to write the result in terms of a difference of integrals of motion $\Delta I_0 \equiv I_0(S) - I_0(J)$. The additional

term $\texttt{res}_\mu$ denotes potential residues that in general will be picked up when shifting the contours in the above maneuvers. In all applications of the Functional SoV method in the existing literature, such residues can be canceled by a judicious choice of $\mu(u)$, which restricts the measure from a generic periodic function to a discrete set of possibilities.

In the simple example of the $\mathfrak{sl}(2)$ spin chain, the restriction of the measure $\mu(u)$ is achieved by requiring that it exponentially decays at infinity (such that the integral against any of the polynomial Q-functions converges), and has at most double poles at $u = \pm i/2$. Such poles are canceled by the double zeros $(u \pm i/2)^2$ of the Baxter operator (2.1) in the potential residue-generating terms, so that the full integrand is free of poles in the region affected by the shifts. The unique measure satisfying these properties is $\mu(u) = \pi/2 \, \text{sech}(\pi u)^2$, which was also used in [22, 26, 36, 44].[4] The relation (2.5) then becomes

$$\Delta I_0 \int_\mathbb{R} Q_S(u) Q_J(u) \mu(u) du = 0. \tag{2.6}$$

Since distinct states have distinct integrals of motion (and conversely, each state is completely specified by its integrals of motions), the integral above must vanish when the two states are different, i.e. we obtain the desired orthogonality relation between Q-functions:

$$\langle Q_S Q_J \rangle_\mu \equiv \int_\mathbb{R} Q_S(u) Q_J(u) \mu(u) du \propto \delta_{S,J} \,. \tag{2.7}$$

Our goal is to obtain a similar orthogonality relation for Q-functions at higher orders in weak-coupling perturbation theory. We focus our attention on the next-to-next-to-leading order,[5] order since it is the first order where almost no results are known for orthogonality,[6] while it still avoids wrapping corrections for the operators. As we will mention below, the game becomes significantly more complicated in the presence of wrapping. Most importantly, the lessons from our $N^2LO$ explorations will serve as guidelines to extend orthogonality to both higher twist and higher orders in perturbation theory.

## 2.2 Higher Loops and Asymptotic Baxter Equation

Beyond the leading order in perturbation theory, the polynomial Q-function above is not sufficient to characterize the spectrum. Instead, the finite-coupling spectrum of conformal primaries in planar $\mathcal{N} = 4$ SYM is encoded in the Quantum Spectral Curve [1, 2]. This is a set of 256 Q-functions $Q_{A|I}(u)$, $A, I \subset \{1, 2, 3, 4\}$, anti-symmetric in both $A$ and $I$, linked by various algebraic and analytic relations. In the asymptotic large-volume regime that we restrict to in this work, a central role is played by the Q-function $Q_{12|12}(u)$. This Q-function encodes the momentum-carrying roots $v_n$, and at leading order is closely related to the polynomial Q-function (2.2) that we considered before. In this regime, it takes the form

$$Q_{12|12}(u) = \mathcal{P}(u)(f^+)^2, \quad \text{with} \quad f^\pm := f(u \pm \tfrac{i}{2}), \tag{2.8}$$

where $\mathcal{P}$ is the polynomial part of the Q-function

$$\mathcal{P}(u) = \prod_{k=1}^{S} (u - v_k). \tag{2.9}$$

---

[4]The most general $i$-periodic measure that decays exponentially at infinity is $\mu(u) = \tanh^m(\pi u) \, \text{sech}^{2n}(\pi u)$ with $m \geq 0$, $n \geq 1$. It has a pole of degree $m + 2n$ at $u = \pm i/2$, hence we require $m + 2n \leq 2$, which leaves $n = 1$, $m = 0$ as the only choice.

[5]We will refrain to use the term "loops", due to the confusion it may cause between the order of corrections in the Q-function and the actual loop order in the field theory, e.g. order $O(g^4)$ in the Q-function computes corrections of order $O(g^6)$ in the spectrum of the gauge theory.

[6]There is a $N^2LO$ orthogonality for twist-2 operators written in [36]. However, the measure used there is not universal but rather depends on the states appearing in the inner product.

The roots $v_k$ are the momentum-carrying roots (which acquire loop corrections in perturbation theory) of the Asymptotic Bethe Ansatz (ABA) equations associated to the middle node of the $\mathfrak{su}(2,2|4)$ Dynkin diagram. In the case of the $\mathfrak{sl}(2)$ sector we consider here, the ABA equations are recalled in Appendix A. Additionally, we have the zero-momentum condition $\mathcal{P}(-i/2) = \mathcal{P}(i/2)$.

The function $f(u)$ is a non-polynomial dressing, with no poles or zeros anywhere, and no branch cuts in the upper half plane [2], that satisfies

$$\frac{f(u+i)}{f(u)} = \prod_{k=1}^{S} \left( \frac{\frac{1}{x(u)} - x_k^+}{\frac{1}{x(u)} - x_k^-} \right) \left( \frac{x_k^-}{x_k^+} \right)^{\frac{1}{2}}, \tag{2.10}$$

where $x(u)$ is the Zhukovsky map (see Appendix A). A solution to this equation, with power-like asymptotics at infinity and analytic in the upper half plane, is given by

$$f(u) = \exp\left( g^2 q_1^+ \psi_0(-iu) + \frac{g^4}{2} \left( i q_2^- \psi_1(-iu) - q_1^+ \psi_2(-iu) \right) + \mathcal{O}(g^6) \right) \tag{2.11}$$

where $\psi_k$ are the polygamma functions of order $k$ and $q_k^\pm$ are conserved charges, also recalled in Appendix A.

To quick-start our SoV explorations, we will consider a slight generalization of the momentum-carrying Q-function, where we symmetrize its dressing, and rescale it by a parameter:

$$Q_S(u) = \prod_{n=1}^{S} (u - v_n) e^{\alpha \cdot \sigma(u)}, \qquad \sigma(u) = \log(f^+ \bar{f}^-), \tag{2.12}$$

where the symmetrized dressing $\sigma(u)$ can be easily computed from (2.10) at any order in perturbation theory. For example, the first few orders are

$$\sigma(u) = 2g^2 q_1^+ \psi_{0,+}(u) - g^4 \left( q_1^+ \psi_{2,+}(u) + q_2^- \psi_{1,-}(u) \right) + \mathcal{O}(g^6), \tag{2.13}$$

where the symmetric and anti-symmetric combinations of polygamma functions are

$$\psi_{n,\pm}(u) = \frac{1}{2} \left( \psi_n \left( \frac{1}{2} + iu \right) \pm \psi_n \left( \frac{1}{2} - iu \right) \right). \tag{2.14}$$

The QSC Q-function $Q_{12|12}(u)$ is analytic in the upper half plane (i.e. its dressing $f^+$ has poles only on the negative imaginary axis, spaced by $i/2$), while the Q-function (2.12) with symmetrized dressing $\sigma$ has poles for both negative and positive imaginary values. Since $f^+/\bar{f}^-$ is a periodic function, when the dressing parameter $\alpha$ is one, the QSC Q-function (2.8) and the symmetrized Q-function (2.12) satisfy the same Baxter equation. For generic values of dressing parameter $\alpha$, this is no longer the case, but it is still easy to write the Baxter equation that the Q-functions (2.12) satisfy:

$$0 = \mathcal{B}_S \cdot Q_S = B_S^+ Q_S(u+i) + B_S^- Q_S(u+i) + T_S(u) Q_S(u), \tag{2.15}$$

where the shift operators are given by[7]

$$B_S^\pm = (x^\pm)^2 \pm i g^2 q_1^+ (1 - 2\alpha)(x^\pm)^1 + g^4 \left( q_2^+ + (q_1^+)^2 (1 - 2\alpha)^2 \right)(x^\pm)^0, \tag{2.16}$$

---

[7]In principle, we could also refer to the conserved charges as integrals of motion, since they are ultimately functions of the state's quantum numbers. However, we will make a distinction between these objects, by denoting as integrals of motion the quantities that appear in the transfer matrix ($T_S$) and as charges the quantities that appear in the shift operators ($B_S^\pm$).

and the transfer matrix is

$$T_S = I_0(S) - 2u^2 + g^4 \left( \frac{1}{(x^-)^2} + \frac{1}{(x^+)^2} \right).$$  (2.17)

Since three different values of $\alpha$ will be important in the following, let us establish some notation before diving into the SoV explorations. Starting from (2.12), we define three Q-functions that are the same at leading order, but differ in perturbation theory:

$$\mathcal{P}_S(u) = Q_S(u)|_{\alpha=0}, \quad \mathcal{Q}_S(u) = Q_S(u)|_{\alpha=1}, \quad \mathbb{Q}_S(u) = Q_S(u)|_{\alpha=\frac{1}{2}}.$$  (2.18)

As mentioned before, when $\alpha = 1$, this is the usual (symmetrized) momentum-carrying QSC Q-function. When $\alpha = 0$, the dressing vanishes and we are left only with the polynomial part of this Q-function. The Q-function with $\alpha = 1/2$, also considered in [36], will be important below and in Section 4, but as yet has no realization in terms of combinations of QSC objects.

In the presence of wrapping, several complications arise. First of all, the form of the Q-functions will become complicated, and will no longer be parametrized by Bethe roots. In principle, one can still compute the Q-functions explicitly at any order at weak coupling, using the package of [45]. However, since we lose the meaning of Bethe roots beyond wrapping order, it is far less clear why a single Q-function should play a prominent role in the SoV expression. It is more natural to expect that the full result involves the entire $\mathfrak{psu}(2,2|4)$ structure, which is far from being understood in functional SoV methods. For these reasons, we focus our attention on the situation where no wrapping effects are present.

## 2.3 Setting up Higher Loop Orthogonality

By simply inspecting the shift operator (2.16), one might be tempted to consider $\alpha = 1/2$, as was done before [36], and in fact, that is exactly what we will initially do. Although it *will not* lead to N²LO orthogonality, it will illustrate the problems one must overcome when considering orthogonality at higher orders in perturbation theory. Simply setting $\alpha = 1/2$ in (2.15) results in the simple Baxter equation

$$\left( (x^+)^2 + g^4 \frac{q_2^+}{2} \right) \mathbb{Q}_S(u+i) + \left( (x^-)^2 + g^4 \frac{q_2^+}{2} \right) \mathbb{Q}_S(u-i) =$$
$$- \left( I_0(S) - 2u^2 + g^4 \left( \frac{1}{(x^-)^2} + \frac{1}{(x^+)^2} \right) \right) \mathbb{Q}_S(u).$$  (2.19)

Note also that the QSC-inspired dressing (2.12) for any value of $\alpha$ has a property that makes the SoV explorations remarkably simple: The non-polynomial part of the transfer matrix is state-independent, while the shift terms remain simple polynomials in $x^\pm$ and conserved charges. Therefore, when we consider differences of Baxter operators as in equation (2.3), these non-polynomial parts will cancel out (just like the universal term $-2u^2$ canceled at leading order). This reduces the number of integrals of motions that appear in the SoV expressions and, as we will see, simplifies the search for orthogonal measures.

Starting from the initial equation (2.3), and performing the same contour manipulations as we did at leading order, we obtain the following N²LO expression:

$$\Delta I_0 \int_{\mathbb{R}} \mathbb{Q}_S(u) \mathbb{Q}_J(u) \mu(u) du + g^4 \frac{\Delta q_2^+}{2} \int_{\mathbb{R}} (B_0 \cdot \mathbb{Q}_S) \mathbb{Q}_J(u) \mu(u) du + \texttt{res}_\mu = 0,$$  (2.20)

where $B_0$ is a particular case of what will be an important new ingredient of orthogonality beyond leading order: The *lower-length Baxter operators*, defined as

$$B_M \cdot F = \left(u + \frac{i}{2}\right)^M F(u+i) + \left(u - \frac{i}{2}\right)^M F(u-i) - 2u^M F(u)\,, \qquad (2.21)$$

whose "transfer matrix" (coefficient of the $F(u)$ term on the r.h.s.) differs from the usual Baxter operator (2.1), since it depends *only* on the universal $u^M$ term. We will comment more on this difference, and the central role of these operators once we consider orthogonality for general twists in Section 3.

In general, to obtain orthogonality relations akin to (2.7), one needs to find as many measures as independent integrals of motion and charges appear in the equation. For example, in the expression (2.20) we have *one* integral of motion $\Delta I_0$ and *one* conserved charge $\Delta q_2^+$ (notice that we have eliminated $q_1^+$ and $(q_1^+)^2$ from the equation by our choice of $\alpha = 1/2$), so *two* measures with vanishing residues are needed to define orthogonality. Let's assume that two such measures $\mu_1(u)$ and $\mu_2(u)$ exist (each intended as an expansion in the coupling up to $O(g^4)$). Then we could write the following linear system

$$\begin{cases} \Delta I_0 \langle \mathbb{Q}_S \mathbb{Q}_J \rangle_{\mu_1} + g^4 \dfrac{\Delta q_2^+}{2} \langle (B_0 \cdot \mathbb{Q}_S) \mathbb{Q}_J \rangle_{\mu_1} = 0 \\[2mm] \Delta I_0 \langle \mathbb{Q}_S \mathbb{Q}_J \rangle_{\mu_2} + g^4 \dfrac{\Delta q_2^+}{2} \langle (B_0 \cdot \mathbb{Q}_S) \mathbb{Q}_J \rangle_{\mu_2} = 0 \end{cases} \qquad (2.22)$$

where we used the notation introduced in (2.7) to denote the integrals of the Q-functions over the real line. It will be useful to rewrite the linear system above as the following matrix equation

$$\begin{pmatrix} \langle \mathbb{Q}_S \mathbb{Q}_J \rangle_{\mu_1} & \langle (B_0 \cdot \mathbb{Q}_S) \mathbb{Q}_J \rangle_{\mu_1} \\ \langle \mathbb{Q}_S \mathbb{Q}_J \rangle_{\mu_2} & \langle (B_0 \cdot \mathbb{Q}_S) \mathbb{Q}_J \rangle_{\mu_2} \end{pmatrix} \cdot \begin{pmatrix} \Delta I_0 \\ g^4/4\, \Delta q_2^+ \end{pmatrix} = 0\,. \qquad (2.23)$$

Since distinct states have non-coincident vectors of integrals of motion, the fact that the homogeneous linear system has a solution implies that the determinant of the matrix of inner products is zero, i.e. it implies the following orthogonality relation for the Q-functions:

$$\begin{vmatrix} \langle \mathbb{Q}_S \mathbb{Q}_J \rangle_{\mu_1} & \langle (B_0 \cdot \mathbb{Q}_S) \mathbb{Q}_J \rangle_{\mu_1} \\ \langle \mathbb{Q}_S \mathbb{Q}_J \rangle_{\mu_2} & \langle (B_0 \cdot \mathbb{Q}_S) \mathbb{Q}_J \rangle_{\mu_2} \end{vmatrix} \propto \delta_{S,J} + \mathcal{O}(g^6)\,. \qquad (2.24)$$

To find the measures with vanishing residues, we follow the method described in Appendix C: We write the ansatz for the measure as a Laurent series, and computes the residues as an explicit combination of the coefficients of this series expansion and the conserved charges of the Q-functions. Since the conserved charges are all independent,[8] demanding that the residues cancel can be done analytically by requiring that the coefficient of each conserved charge vanishes. This process fixes all the coefficients of the Laurent series of the measure, which in turn determines the sought-after measures with vanishing residues.

It turns out that, for the particularly simple case we have been considering so far (Q-functions with dressing parameter $\alpha = 1/2$), and in fact for any value of dressing parameter $\alpha$, this analytic residue computation shows that only *one* measures with vanishing residues exists (which we describe in more details in Section 4). Since we do not have as many measures with vanishing residues as integrals of motion and conserved charges, it seems that N²LO orthogonality for twist-two operators, if it exists, must have different ingredients than its previous orders in perturbation theory.

---

[8]For a fixed twist, all conserved charges are linearly independent functions of the spin. For example, for twist-two operators at leading order, we have: $q_1^+ = 4H_1(S)$ and $q_2^+ = 2\left(H_2\left(\frac{s-1}{2}\right) - H_2\left(\frac{s}{2}\right) + \frac{\pi^2}{3}\right)$, where $H_n(S)$ are the harmonic numbers.

## 2.4 Exploring Twist-Three Orthogonality

To better understand what types of structures we can expect at higher loops, let us consider twist-three operators at one loop.

The main new technical obstacle that arises for twist three is that, unlike at twist two, not all states are parity-symmetric. Non-symmetric states start to appear at spins $S \geq 6$ and these states have a non-zero charge $q_1^-$. This charge enters the coefficients of the Baxter operator at one loop

$$\mathcal{B} \cdot \mathbb{Q} = (x^+)^3 \left(1 + g^2 \frac{q_1^-}{x^+}\right) \mathbb{Q}(u+i) + (x^-)^3 \left(1 + g^2 \frac{q_1^-}{x^-}\right) \mathbb{Q}(u-i) - T(u)\mathbb{Q}(u) = 0 \quad (2.25)$$

which annihilates the dressed Q-function $\mathbb{Q}(u)$ with $\alpha = 1/2$ considered in the previous section. The transfer matrix is given by

$$T(u) = 2u^3 - 2g^2 q_1^- + I_1 u + I_0 \,, \quad (2.26)$$

and we remind that $I_1$ and $I_0$ are functions that depend both on the state and on the coupling.

Note that from a practical perspective, the obstacles posed by the presence of $q_1^-$ at one loop are very similar to those of $q_2^+$ at two loops for twist-two, as both enter the Baxter equation in an almost identical fashion.

Since $q_1^-$ always appears with a factor of $g^2$, it does not contribute at leading order. Then we only need to find two independent measures such that the residues vanish. Following the logic for twist-two, these measures can be computed to be given by

$$\mu_1(u) = \frac{\pi}{2 \cosh^2(\pi u)}, \quad \mu_2(u) = \frac{\pi^2 \tanh(\pi u)}{\cosh^2(\pi u)} \,. \quad (2.27)$$

This results in the leading-order orthogonality relation

$$\begin{pmatrix} \langle \mathcal{P}_S \mathcal{P}_J \rangle_{\mu_1} & \langle \mathcal{P}_S \, u \, \mathcal{P}_J \rangle_{\mu_1} \\ \langle \mathcal{P}_S \mathcal{P}_J \rangle_{\mu_2} & \langle \mathcal{P}_S \, u \, \mathcal{P}_J \rangle_{\mu_2} \end{pmatrix} \cdot \begin{pmatrix} \Delta I_0 \\ \Delta I_1 \end{pmatrix} = 0 \,. \quad (2.28)$$

We now proceed exactly as we did for twist two, and try to find measures such that the residues cancel. In this case, we need to find three measures $\mu_i$, since we have three non-trivial integrals of motion in the Baxter equation at this loop order: Two integrals of motion $I_0$ and $I_1$ coming from the transfer matrix, as well as $q_1^-$. At leading order, we can indeed find three such measures $\mu_\ell$, $\ell = 1, 2, 3$, *provided* we restrict our analysis to zero-momentum states. These measures are given explicitly by

$$\mu_\ell(u) = \frac{\ell \pi^\ell / 2}{\cosh^2(\pi u)} \tanh^{\ell-1}(\pi u) \,, \quad (2.29)$$

where the overall normalization is rather unimportant. Indeed, for $\mu_3$ the residue contribution at leading order is proportional to

$$\frac{\mathcal{P}_S(-i/2)}{\mathcal{P}_S(i/2)} - \frac{\mathcal{P}_J(-i/2)}{\mathcal{P}_J(i/2)} \,, \quad (2.30)$$

and so it vanishes when both states satisfy the zero-momentum condition $\mathcal{P}(-i/2) = \mathcal{P}(i/2)$.

Proceeding to the first order in the weak-coupling expansion, we find that by considering simple perturbative corrections to the measures (2.29), given by

$$\mu_\ell(u) = \frac{\ell \pi^\ell / 2}{\cosh^2(\pi u)} \tanh^{\ell-1}(\pi u) \left(1 + g^2 \pi^2 \left(-\frac{\ell}{3} + (\ell+2) \tanh^2(\pi u)\right)\right) + \mathcal{O}(g^4) \,, \quad (2.31)$$

we can make the residues vanish at NLO, provided again that the states satisfy the zero-momentum condition. Hence, we obtain the following linear system:

$$\begin{pmatrix} \langle \mathbb{Q}_S \mathbb{Q}_J \rangle_{\mu_1} & \langle \mathbb{Q}_S\, u\, \mathbb{Q}_J \rangle_{\mu_1} & \langle (B_2 \cdot \mathbb{Q}_S)\mathbb{Q}_J \rangle_{\mu_1} \\ \langle \mathbb{Q}_S \mathbb{Q}_J \rangle_{\mu_2} & \langle \mathbb{Q}_S\, u\, \mathbb{Q}_J \rangle_{\mu_2} & \langle (B_2 \cdot \mathbb{Q}_S)\mathbb{Q}_J \rangle_{\mu_2} \\ \langle \mathbb{Q}_S \mathbb{Q}_J \rangle_{\mu_3} & \langle \mathbb{Q}_S\, u\, \mathbb{Q}_J \rangle_{\mu_3} & \langle (B_2 \cdot \mathbb{Q}_S)\mathbb{Q}_J \rangle_{\mu_3} \end{pmatrix} \begin{pmatrix} \Delta I_0 \\ \Delta I_1 \\ g^2 \Delta q_1^- \end{pmatrix} = 0 \qquad (2.32)$$

where $B_2$ is the "lower-length" Baxter operator of (2.21):

$$B_2 \cdot F = \left(u + \frac{i}{2}\right)^2 F(u+i) + \left(u - \frac{i}{2}\right)^2 F(u-i) - 2u^2 F(u) \qquad (2.33)$$

as usual, the determinant of the coefficient matrix must vanish for two distinct states, and thus provides an NLO orthogonality relation for twist-three states. This construction can be extended to any twist $L$, and takes the form of an $L \times L$ determinant. Interestingly, this seems to provide a different one-loop orthogonality relation than found in [36], which takes the form of a $(L-1) \times (L-1)$ matrix. The relation between the two is not completely clear, as the relation found in that paper was not systematically derived.

Unfortunately, we were not able to extend this observation to higher loops. The main issue we encounter, which is a general obstacle not restricted to the present set-up, is that at two loops, more integrals of motion will appear in the equation, which necessitates more measures. Any additional measure we introduce does not have vanishing residues, even at tree level, and these residues are proportional to yet more integrals of motion, necessitating even more measures, and so on. In short, there is an uncontrolled runaway of integrals of motion, even at fixed orders of perturbation theory.

Let us note that the observations above may be a hint that for higher loops we need to consider more general solutions of the Baxter equation. Indeed, the construction above crucially relies on the multiplication of the powerlike solution by the periodic function defined above (2.15), and the residues in the system of equations does not vanish without it. In principle, we can also involve the second (non-polynomial) solution of the Baxter equation, and we will do exactly that in Section 4.2. While this Q-function may seem unnatural at weak coupling, at finite coupling, all Q-functions are essentially on equal footing. Understanding exactly which solutions of the Baxter equation are needed for orthogonality at higher loops may provide valuable insights into a finite-coupling SoV construction.

Before describing explorations in this direction in Section 4.2, we consider other possible ways to widen the usual methodology of functional SoV. We will use some of these ideas, abandoning some of the usual assumptions, to directly try to construct bilinear integrals of Q-functions with vanishing determinant for two distinct states, which is the end goal of the construction anyway.

## 2.5   What Could be Done Differently?

Even though we encounter the roadblock of not finding enough measures with vanishing residues at N²LO (both at twist two and at twist three), there are still several alternatives to be considered in the Separation of Variables formalism.

One possibility is to radically change the approach, and consider a different integration contour, such as a vertical contour parallel to the imaginary axis, as was employed to obtain SoV-like expressions for certain two-point and three-point functions in the context of cusped Wilson loops in the ladders limit [32], or in the fishnet theory [35, 46]. The main advantage of such contours is that in all manipulations needed to prove orthogonality, the contour shifts onto itself, so there would be no residues to pick. However, to guarantee convergence with

such integration contours, it is necessary to deform the system by twisting the boundary conditions of the Q-functions, which modifies their asymptotics [26, 28, 46]. In this work, we will not twist the Q-functions, and thus we will not follow this route. Instead, we will push the results of integration along the horizontal line beyond leading-order perturbation theory.

The way to do that is to embrace the fact that the residues will not be zero anymore. This will shift our focus from demanding that measures have vanishing residues to demanding that the residues have a very specific form. This will allow us to construct linear systems akin to (2.23), and will result in orthogonality relations among the Q-functions. To obtain such orthogonality relations, we will follow two distinct approaches, one more analytic in nature, while the other is based on numerical explorations.

The more analytic approach is to enlarge the system by considering the discarded second solution to the Baxter equation. The motivation behind this is to treat the rank-one $\mathfrak{sl}(2)$ sector in the same way as the higher-rank sectors [26, 28, 43], whose orthogonality is given by a combination of all the different types of Q-functions of the system. Considering both Q-functions of the $\mathfrak{sl}(2)$ sector, together with allowing for non-vanishing residues, will result in the N²LO orthogonality for twist-two operators presented in Section 4. However, this approach becomes rapidly very complicated, hence we were not able to generalize it to operators with arbitrary twist at higher orders in perturbation theory.

Another more experimental approach we took is to forget the different types of Q-functions, the initial equation (2.1) and its residues, and just search numerically for measures and enlarged matrices of Q-functions, like the one proposed in (2.24), that give orthogonality beyond leading order. To construct a sensible ansatz for these putative enlarged matrices, we will draw inspiration from the twist-three case at NLO discussed above, where we have complete analytic control, and where we have indeed found such a matrix equation (2.32). Building upon the enlarged matrices we encounter in this setup, we will be able to define orthogonality not only for twist-two operators at N²LO, but also for operators of arbitrary twist at any order in perturbation theory, up to wrapping order (with the aforementioned drawback about degenerate states). This result will also reveal the interesting fact that the dressing parameter $\alpha$ needs to appear asymmetrically for the two states. This is what we will turn to now.

## 2.6 Higher-Loop Measures

Our strategy is to construct putative enlarged matrices similar to (2.23) and (2.32) for twist-two operators, and then search for measures that make their determinant orthogonal at N²LO. To start these explorations, we must first establish how these matrices are constructed. They will obey three simple rules:

**Rule 1.** All Q-functions are of the form (2.12). This fixes the functional form of the dressing to be (2.10), while allowing each Q-function to have its own value of dressing parameter $\alpha$. For any value of $\alpha$, the non-polynomial part of the transfer matrix is universal, as in (2.17). Such state-independent terms cancel in the differences of the initial equation (2.1), and thus minimize the number of integrals of motion that need to be considered.

**Rule 2.** The matrix elements can be either Q-functions $\langle Q_S Q_J \rangle$, or Q-functions modified by lower-length Baxter operators $\langle (B_M \cdot Q_S) Q_J \rangle$ (2.21), as motivated from the twist-three expression (2.32). The length $M$ of these lower-length Baxter operators is dictated by the powers of $x(u)$ that appear in the perturbative expansion of the shift operator. As explicitly seem in the three terms of (2.16): At LO, no lower-length Baxter operators

must be considered, at NLO, the operator $B_1$ can appear, and N²LO the operator $B_0$ can appear.

**Rule 3.** There is no explicit coupling dependence appearing in the matrix elements. This is the simplest setup where orthogonality can arise. To see this, one can consider the case (2.22) with vanishing residues. There are three distinct ways of writing this linear system as a matrix equation. The first way is (2.23), while the other two ways are given by

$$\begin{pmatrix} \langle Q_S Q_J \rangle_{\mu_1} & g^2 \langle B_0[Q_S, Q_J] \rangle_{\mu_1} \\ \langle Q_S Q_J \rangle_{\mu_2} & g^2 \langle B_0[Q_S, Q_J] \rangle_{\mu_2} \end{pmatrix} \cdot \begin{pmatrix} \Delta I_0 \\ \frac{g^2 \Delta q_2^+}{4} \end{pmatrix}, \tag{2.34}$$

or

$$\begin{pmatrix} \langle Q_S Q_J \rangle_{\mu_1} & g^4 \langle B_0[Q_S, Q_J] \rangle_{\mu_1} \\ \langle Q_S Q_J \rangle_{\mu_2} & g^4 \langle B_0[Q_S, Q_J] \rangle_{\mu_2} \end{pmatrix} \cdot \begin{pmatrix} \Delta I_0 \\ \frac{\Delta q_2^+}{4} \end{pmatrix}. \tag{2.35}$$

Due to the explicit coupling dependence of the matrix elements, the determinants of the two matrices above are always zero at leading order, both for equal and for distinct states, which violates leading-order orthogonality for the Q-functions.

Following these rules, we construct the following matrices:

$$\mathcal{M}_{2,0}[Q_S, Q_J] = \begin{pmatrix} \langle Q_S Q_J \rangle_{\mu_1} \end{pmatrix},$$

$$\mathcal{M}_{2,1}[Q_S, Q_J] = \begin{pmatrix} \langle Q_S Q_J \rangle_{\mu_1} & \langle (B_1 \cdot Q_S) Q_J \rangle_{\mu_1} \\ \langle Q_S Q_J \rangle_{\mu_2} & \langle (B_1 \cdot Q_S) Q_J \rangle_{\mu_2} \end{pmatrix},$$

$$\mathcal{M}_{2,2}[Q_S, Q_J] = \begin{pmatrix} \langle Q_S Q_J \rangle_{\mu_1} & \langle (B_1 \cdot Q_S) Q_J \rangle_{\mu_1} & \langle (B_0 \cdot Q_S) Q_J \rangle_{\mu_1} \\ \langle Q_S Q_J \rangle_{\mu_2} & \langle (B_1 \cdot Q_S) Q_J \rangle_{\mu_2} & \langle (B_0 \cdot Q_S) Q_J \rangle_{\mu_2} \\ \langle Q_S Q_J \rangle_{\mu_3} & \langle (B_1 \cdot Q_S) Q_J \rangle_{\mu_3} & \langle (B_0 \cdot Q_S) Q_J \rangle_{\mu_3} \end{pmatrix}. \tag{2.36}$$

where we allow each of the Q-functions above to have its own value of dressing parameter $\alpha$. Our goal is to find out whether there is a combination of dressing parameters $\alpha$ and measures $\mu_i(u)$ that make these enlarged matrices orthogonal.

The first matrix $\mathcal{M}_{2,0}$ has the same form as the one previously considered (2.7), while the other two *enlarged matrices* are new candidates for orthogonality. In general, we will use $\mathcal{M}_{L,\ell}$ to denote the matrix of twist-$L$ operators enlarged $\ell$ times. The orthogonality relation will be computed by the symmetrized determinant

$$\mathcal{D}_{L,\ell}[Q_S, Q_J] = \sqrt{\det \mathcal{M}_{L,\ell}[Q_S, Q_J] \det \mathcal{M}_{L,\ell}[Q_J, Q_S]}, \tag{2.37}$$

where taking the square root of the product is a simple way of recovering the symmetry of exchanging the states $S \leftrightarrow J$, which was broken by the enlarged matrices (2.36) that apply the lower-length Baxter operator to just one of the two states.

**Leading Order.** The fact that the coupling does not appear explicitly in the matrix elements (2.36) means that for $\ell > 0$, even leading-order orthogonality depends non-trivially on the lower-length Baxter operators. Thus, beyond the previously discussed case of $\mathcal{M}_{2,0}$, even at leading order, we have no guarantee that measures that make these enlarged matrices $\mathcal{M}_{2,1}$ and $\mathcal{M}_{2,2}$ orthogonal exist. However, it turns out that such measures do exist. They are exactly the ones considered in the twist-three case (2.29) and are given by

$$\mu_1^{(0)}(u) = \frac{\pi}{2} \frac{1}{\cosh^2(\pi u)}, \quad \mu_2^{(0)}(u) = \pi^2 \frac{\tanh(\pi u)}{\cosh^2(\pi u)}, \quad \text{and} \quad \mu_3^{(0)}(u) = \frac{3\pi^3}{2} \frac{\tanh^2(\pi u)}{\cosh^2(\pi u)}. \tag{2.38}$$

This results in three distinct leading-order orthogonality relations for twist-two operators. The first one, $\mathcal{D}_{2,0}$, is exactly the one considered in (2.7), while the other ($\mathcal{D}_{2,1}$ and $\mathcal{D}_{2,2}$) are two new relations that from now on we will refer to as *enlarged orthogonality*. For example, in the case of the $\ell = 2$ enlarged matrices we have

$$\mathcal{M}_{2,2}[Q_4, Q_4] = \begin{pmatrix} \frac{1}{9} & 0 & 0 \\ 0 & \frac{25}{3} & 0 \\ 3 & 0 & \frac{65}{4} \end{pmatrix},$$

$$\mathcal{M}_{2,2}[Q_4, Q_6] = \begin{pmatrix} 0 & 0 & 0 \\ 0 & \frac{37}{25} & 0 \\ \frac{1}{5} & 0 & \frac{343}{36} \end{pmatrix}, \qquad \mathcal{M}_{2,2}[Q_6, Q_4] = \begin{pmatrix} 0 & 0 & \frac{22}{15} \\ 0 & \frac{7}{15} & 0 \\ \frac{1}{5} & 0 & 17.781 \end{pmatrix},$$

whose determinants are

$$\det(\mathcal{M}_{2,2}[Q_4, Q_4]) = \frac{1625}{108}, \quad \det(\mathcal{M}_{2,2}[Q_4, Q_6]) = 0, \quad \det(\mathcal{M}_{2,2}[Q_6, Q_4]) = -\frac{154}{1125}.$$

While the symmetrized determinant we defined (2.37) has the desired orthogonality property: $\mathcal{D}_{2,2}[Q_S, Q_J] \propto \delta_{S,J}$, we see that the matrices themselves depend on the ordering of the states. More precisely, we find that $\det(\mathcal{M}_{2,2}[Q_S, Q_J]) = 0$ if $S < J$. This ordering of the states emerges from integrals involving lower-length Baxter operators. For example, in the case of the two-times enlarged matrix discussed above, one of these integrals evaluates to

$$\langle (B_0 \cdot Q_S) Q_J \rangle_{\mu_1} = \begin{cases} 0 & \text{for } S > J, \\ 1/(2S+1) & \text{for } S = J, \\ \frac{1}{2}(-1)^{(S-J)/2} \left( q_1^+(J) - q_1^+(S) \right) & \text{for } S < J, \end{cases} \qquad (2.39)$$

where we can clearly see the ordering in the states appearing. This is another reason why we will always use the symmetrized determinant $\mathcal{D}_{L,\ell}$ defined (2.37) to compute orthogonality among the Q-functions.

Moreover, at this order, it is easy to compare these SoV enlarged determinants with more usual quantities, such as the Gaudin norm. We find that these quantities are related by the following normalization factors

$$\mathcal{D}_{2,0}[Q_S, Q_J] = \delta_{S,J} \frac{(2S+1)}{(2S)!} \left( Q_S \left( \tfrac{i}{2} \right) Q_S \left( -\tfrac{i}{2} \right) \right)^{-2} \times \mathbb{B}_2(S),$$

$$\mathcal{D}_{2,1}[Q_S, Q_J] = \delta_{S,J} \frac{(2S+1)}{(2S)!} \left( Q_S \left( \tfrac{i}{2} \right) Q_S \left( -\tfrac{i}{2} \right) \right)^{-3} \times \frac{\mathbb{B}_2(S)}{q_1^+(S)},$$

$$\mathcal{D}_{2,2}[Q_S, Q_J] = \delta_{S,J} \frac{(2S+1)}{(2S)!} \left( Q_S \left( \tfrac{i}{2} \right) Q_S \left( -\tfrac{i}{2} \right) \right)^{-4} \times \frac{\mathbb{B}_2(S)}{q_1^+(S)(q_3^+(S) - 16H_3(S))}, \qquad (2.40)$$

where $H_3(S)$ is the harmonic number and $\mathbb{B}_2(S)$ is the twist-two Gaudin norm defined in Appendix B. There, we also present the generalization of these relations between SoV and Gaudin norm for operators with arbitrary twists. Note that the relations (2.40) are not normalization-invariant. If one rescales the Q-function (2.12) by an overall constant $\mathcal{N}$, the relation with the Gaudin norm at leading order changes by a factor $\mathcal{N}^{L+\ell}$. One can use this dependence to make the expressions above nicer, but it does not change the main feature we want to emphasize: Each enlarged matrix is related to the Gaudin norm by a different normalization factor.

**Higher Orders.** With the leading order established, we can now make precise how these enlarged matrices are used to define N²LO orthogonality at twist $L = 2$. The strategy remains the same: Find perturbative corrections to the measures (2.38) that result in orthogonality beyond leading order. To explore the vast space of possible measures, we assume that the corrections in perturbation theory will have the same form as in (2.31) and [36], which allows us to parametrize them in terms of a handful of constants:

$$\mu_\ell^{(2)}(u) = \mu_\ell^{(0)}(u) \left( 1 + g^2 \pi^2 a_\ell \tanh^2(\pi u) + g^4 \pi^4 (b_\ell \tanh^2(\pi u) + c_\ell \tanh^4(\pi u)) \right) . \quad (2.41)$$

In the end, all the putative enlarged matrices (2.36) combined have 37 unknown coefficients: Nine $\{a_1, \ldots, c_3\}$ that control the perturbative corrections (2.41) of the three measures (2.38), and 28 dressing parameters $\alpha$ for the Q-functions appearing in the matrix elements of (2.36). By requiring orthogonality, i.e. demanding that the determinant of two distinct states vanishes, we were able to uniquely[9] fix all these coefficients.[10]

The result can be compactly written as a set of three enlarged matrices:

$$\mathcal{M}_{2,0}[Q_S, Q_J] = \left( \langle \mathcal{P}_S \mathcal{P}_J \rangle_{\mu_1} \right) ,$$

$$\mathcal{M}_{2,1}[Q_S, Q_J] = \begin{pmatrix} \langle \mathcal{P}_S \mathcal{Q}_J \rangle_{\mu_1} & \langle (B_1 \cdot \mathcal{P}_S) \mathcal{Q}_J \rangle_{\mu_1} \\ \langle \mathcal{P}_S \mathcal{Q}_J \rangle_{\mu_2} & \langle (B_1 \cdot \mathcal{P}_S) \mathcal{Q}_J \rangle_{\mu_2} \end{pmatrix} ,$$

$$\mathcal{M}_{2,2}[Q_S, Q_J] = \begin{pmatrix} \langle \mathcal{P}_S \mathcal{Q}_J \rangle_{\mu_1} & \langle (B_1 \cdot \mathcal{P}_S) \mathcal{Q}_J \rangle_{\mu_1} & \langle (B_0 \cdot \mathcal{P}_S) \mathcal{Q}_J \rangle_{\mu_1} \\ \langle \mathcal{P}_S \mathcal{Q}_J \rangle_{\mu_2} & \langle (B_1 \cdot \mathcal{P}_S) \mathcal{Q}_J \rangle_{\mu_2} & \langle (B_0 \cdot \mathcal{P}_S) \mathcal{Q}_J \rangle_{\mu_2} \\ \langle \mathcal{P}_S \mathcal{Q}_J \rangle_{\mu_3} & \langle (B_1 \cdot \mathcal{P}_S) \mathcal{Q}_J \rangle_{\mu_3} & \langle (B_0 \cdot \mathcal{P}_S) \mathcal{Q}_J \rangle_{\mu_3} \end{pmatrix} . \quad (2.42)$$

Recall that $\mathcal{Q}_S(u)$ denotes Q-functions with dressing parameter $\alpha = 1$ (these are exactly the symmetrized momentum-carrying Q-function of the QSC), while $\mathcal{P}_S(u)$ have $\alpha = 0$, i.e. they are the polynomial parts of the corresponding $\mathcal{Q}_S(u)$. Meanwhile, the corrected measures in perturbation theory are given by

$$\mu_1^{(2)}(u) = \frac{\pi}{2} \frac{1}{\cosh^2(\pi u)} \left( 1 + 2\pi^2 g^2 \tanh^2(\pi u) - \frac{g^4 \pi^4}{3} \left( 11 \tanh^2(\pi u) - 15 \tanh^4(\pi u) \right) \right) ,$$

$$\mu_2^{(2)}(u) = \pi^2 \frac{\tanh(\pi u)}{\cosh^2(\pi u)} \left( 1 + 3\pi^2 g^2 \tanh^2(\pi u) - \frac{g^4 \pi^4}{3} \left( 18 \tanh^2(\pi u) - 27 \tanh^4(\pi u) \right) \right) ,$$

$$\mu_3^{(2)}(u) = \frac{3\pi^3}{2} \frac{\tanh^2(\pi u)}{\cosh^2(\pi u)} \left( 1 + 4\pi^2 g^2 \tanh^2(\pi u) - \frac{g^4 \pi^4}{3} \left( 25 \tanh^2(\pi u) - 42 \tanh^4(\pi u) \right) \right) . $$
$$(2.43)$$

With the dressing and measures fixed, we can now make the precise statement of enlarged orthogonality for twist-two operators: The matrix enlarged $\ell$ times defines orthogonality up to order N$^\ell$LO, or equivalently

$$\mathcal{D}_{2,0}[Q_S, Q_J] \propto \delta_{S,J} + O(g^2) ,$$
$$\mathcal{D}_{2,1}[Q_S, Q_J] \propto \delta_{S,J} + O(g^4) ,$$
$$\mathcal{D}_{2,2}[Q_S, Q_J] \propto \delta_{S,J} + O(g^6) . \quad (2.44)$$

---

[9]The measures are fixed up to constant terms, which can be neglected, since at each order in perturbation theory they will multiply factors that are vanishing due to the orthogonality relations of the previous perturbative orders.

[10]In practice, the coefficients are fixed by demanding orthogonality among all the states up to spin $S = 10$. Then we verified that these coefficients still imply orthogonality between states up to spin $S = 30$.

The two-times-enlarged determinant $\mathcal{D}_{2,2}$ is our new SoV-inspired proposal of N²LO orthogonality for twist-two operators. For example, when the two operators have spins four and six, this two-times-enlarged matrix can be easily computed numerically to be[11]

$$
\det \mathcal{M}_{2,2}[Q_4, Q_6] =
\begin{vmatrix}
3.2\, g^4 & -12.1i\, g^2 + 220.3i\, g^4 & 152.5\, g^4 \\
-1.3i\, g^2 - 4i\, g^4 & -2.5 + 10.4\, g^2 + 820.7\, g^4 & -62.2i\, g^2 + 873.4\, g^4 \\
-0.2 - 3.4\, g^2 + 113.9\, g^4 & -304.3i\, g^2 - 830.7i\, g^4 & -9.5 + 0.2\, g^2 + 4020.3\, g^4
\end{vmatrix} = \mathcal{O}(g^6)\,.
$$

The fact that the matrices responsible for orthogonality grow with the order in perturbation theory is consistent with the SoV expectations. However, contrary to expectations, the enlargement we are obtaining does not seem to come from a weak-coupling reduction of some larger matrix (at least we were not able to write it like that). If a simple perturbative reduction existed, the normalization factor between the Gaudin norm and each one of the enlarged determinants ($\mathcal{D}_{2,0}$, $\mathcal{D}_{2,1}$ and $\mathcal{D}_{2,2}$) should be the same at leading order, and only start to differ as we go to higher orders in perturbation theory. Instead, we observe that the enlarged matrices have increasingly more complicated normalization factors (2.40) with the Gaudin norm already at leading order.

Due to this increase in complexity, we were not able to write the precise relation between our SoV expressions and the Gaudin norm beyond leading order, or equivalently, the precise normalization between SoV determinants and two-point functions. The best we were able to do, for twist-two as well as for higher-twist operators, is to find the cases where this two-point function vanishes. More precisely, we were able to find matrices and measures that define orthogonality relations for the Q-functions at each order in perturbation theory up to wrapping corrections. This is what we turn to next.

# 3  Orthogonality Before Wrapping

## 3.1  Proposal

Before considering the weak-coupling expansion for operators of general twist, let us first recall the Baxter equation for operators of twist $L$ at leading order:

$$
\left(u + \frac{i}{2}\right)^L Q_S(u+i) + \left(u - \frac{i}{2}\right)^L Q_S(u-i) - \left(2u^L + \sum_{k=0}^{L-2} I_k(S)u^k\right) Q_S(u) = 0\,. \tag{3.1}
$$

Due to the appearance of $(L-1)$ integrals of motion $I_k(S)$, the orthogonality for higher-twist operators is given by a matrix already at leading order. Orthogonality of the Q-functions is the statement that for any two distinct twist-$L$ states, the determinant of the following $(L-1) \times (L-1)$ matrix vanishes:

$$
\mathcal{M}_{L,0}[Q_S, Q_J] =
\begin{pmatrix}
\langle \mathcal{P}_S \mathcal{P}_J \rangle_1 & \langle u^1 \mathcal{P}_S \mathcal{P}_J \rangle_1 & \dots & \langle u^{L-2} \mathcal{P}_S \mathcal{P}_J \rangle_1 \\
\langle \mathcal{P}_S \mathcal{P}_J \rangle_2 & \langle u^1 \mathcal{P}_S \mathcal{P}_J \rangle_2 & \dots & \langle u^{L-2} \mathcal{P}_S \mathcal{P}_J \rangle_2 \\
\vdots & \vdots & \vdots & \vdots \\
\langle \mathcal{P}_S \mathcal{P}_J \rangle_{L-1} & \langle u^1 \mathcal{P}_S \mathcal{P}_J \rangle_{L-1} & \dots & \langle u^{L-2} \mathcal{P}_S \mathcal{P}_J \rangle_{L-1}
\end{pmatrix}, \tag{3.2}
$$

---

[11]One might not be impressed by the N²LO orthogonality exemplified, since some matrix elements are already of order $\mathcal{O}(g^4)$, but this is an artifact of twist-two operators and their symmetric Bethe roots. When we promote orthogonality to operators with arbitrary twist, all matrix elements will generically start at order $\mathcal{O}(g^0)$.

where we emphasize the leading-order aspect of this result by writing only the polynomial part $\mathcal{P}_S$ of the Q-functions (2.9), and recall that each of the brackets above stands for the integral on the real line

$$\langle f \rangle_\ell = \int_{-\infty}^{\infty} f(u)\mu_\ell(u)\,du\,, \tag{3.3}$$

where the measures at leading order are given by

$$\mu_\ell^{(0)}(u) = \frac{\ell\,\pi^\ell/2}{\cosh^2(\pi u)}\tanh^{\ell-1}(\pi u)\,. \tag{3.4}$$

To promote this result beyond leading order, for each matrix $\mathcal{M}_{L,0}$ and integer $0 \leq \ell \leq L$, we introduce the $\ell$-times enlarged matrix

$$\mathcal{M}_{L,\ell}[Q_S, Q_J] = \tag{3.5}$$
$$\begin{pmatrix} \langle \mathcal{P}_S\mathcal{Q}_J \rangle_1 & \cdots & \langle u^{L-2}\mathcal{P}_S\mathcal{Q}_J \rangle_1 & \langle (B_{L-1}\mathcal{P}_S)\mathcal{Q}_J \rangle_1 & \cdots & \langle (B_{L-\ell}\mathcal{P}_S)\mathcal{Q}_J \rangle_1 \\ \langle \mathcal{P}_S\mathcal{Q}_J \rangle_2 & \cdots & \langle u^{L-2}\mathcal{P}_S\mathcal{Q}_J \rangle_2 & \langle (B_{L-1}\mathcal{P}_S)\mathcal{Q}_J \rangle_2 & \cdots & \langle (B_{L-\ell}\mathcal{P}_S)\mathcal{Q}_J \rangle_2 \\ \vdots & \vdots & \vdots & \vdots & \vdots & \vdots \\ \langle \mathcal{P}_S\mathcal{Q}_J \rangle_{L-1} & \cdots & \langle u^{L-2}\mathcal{P}_S\mathcal{Q}_J \rangle_{L-1} & \langle (B_{L-1}\mathcal{P}_S)\mathcal{Q}_J \rangle_{L-1} & \cdots & \langle (B_{L-\ell}\mathcal{P}_S)\mathcal{Q}_J \rangle_{L-1} \\ \langle \mathcal{P}_S\mathcal{Q}_J \rangle_L & \cdots & \langle u^{L-2}\mathcal{P}_S\mathcal{Q}_J \rangle_L & \langle (B_{L-1}\mathcal{P}_S)\mathcal{Q}_J \rangle_L & \cdots & \langle (B_{L-\ell}\mathcal{P}_S)\mathcal{Q}_J \rangle_L \\ \vdots & \vdots & \vdots & \vdots & \vdots & \vdots \\ \langle \mathcal{P}_S\mathcal{Q}_J \rangle_{L-1+\ell} & \cdots & \langle u^{L-2}\mathcal{P}_S\mathcal{Q}_J \rangle_{L-1+\ell} & \langle (B_{L-1}\mathcal{P}_S)\mathcal{Q}_J \rangle_{L-1+\ell} & \cdots & \langle (B_{L-\ell}\mathcal{P}_S)\mathcal{Q}_J \rangle_{L-1+\ell} \end{pmatrix},$$

where we remind that $\mathcal{P}_S$ and $\mathcal{Q}_S$ are given by (2.12) with dressing parameters $\alpha = 0$ and $\alpha = 1$, respectively, while $B_M$ stands for the lower-length Baxter operator

$$B_M\mathcal{P}_S = \left(u + \frac{i}{2}\right)^M \mathcal{P}_S(u+i) + \left(u - \frac{i}{2}\right)^M \mathcal{P}_S(u-i) - 2u^M\mathcal{P}_S(u)\,. \tag{3.6}$$

The upper left $(L-1) \times (L-1)$ part (black) is the usual orthogonality matrix, while the new rows and columns (red) are the enlargement to a $(L-1+\ell) \times (L-1+\ell)$ matrix via the addition of lower-length Baxter operators and further measures.

Our main result can be stated in the following way:

The enlarged matrices $\mathcal{M}_{L,\ell}$ define twist-$L$ orthogonality at N$^\ell$LO :

$$\mathcal{D}_{L,\ell}[Q_S, Q_J] \equiv \sqrt{\det \mathcal{M}_{L,\ell}[Q_S, Q_J]\det \mathcal{M}_{L,\ell}[Q_J, Q_S]} \propto \delta_{S,J} + \mathcal{O}(g^{2(\ell+1)}) \tag{3.7}$$

Where the measures responsible for this orthogonality are given by

$$\mu_\ell(u) = \frac{\ell\pi^\ell/2}{\cosh^2(\pi u)}\tanh^{\ell-1}(\pi u)\,e^{\nu_\ell(u)}\,, \tag{3.8}$$

$$\nu_\ell(u) = \sum_{n=1}^{\infty} \left(\binom{2n-1}{n}\frac{\ell+1}{n} - (g\pi)^2\binom{2n}{n}\frac{(\ell+1)(1+4n/3)-1}{n+1}\right)(g\pi)^{2n}\tanh^{2n}(\pi u)\,.$$

We emphasize that the measures $\mu_l$ above are twist-independent, depending only on the row of the matrix in which they appear. Note also that just as in the twist-two case, the enlarged matrices will be sensitive to the ordering of the states: $\det(\mathcal{M}_{L,\ell}[Q_S, Q_J]) = 0$ only if $S < J$, thus it is necessary to consider the symmetrized determinant $\mathcal{D}_{L,\ell}$ to define the

orthogonality (3.7) among the Q-functions. Note also that for any ordering of the states, when one evaluates the determinant of the enlarged matrices (3.5), it is possible to factor out the measures and Q-functions to write the orthogonality relation in a very suggestive form:

$$\det(\mathcal{M}_{L,\ell}) = \int \left( \prod_{n=1}^{L+\ell-1} du_n \, \mu_n(u_n) \right) \times \left( \prod_{n=1}^{L+\ell-1} \mathcal{P}_S(u_n) \right) \times \overleftarrow{\mathbb{V}} \times \left( \prod_{n=1}^{L+\ell-1} \mathcal{Q}_J(u_n) \right), \quad (3.9)$$

where $\overleftarrow{\mathbb{V}}$ is an operator that acts on the Q-functions $\mathcal{P}_S$ on the left defined as

$$\overleftarrow{\mathbb{V}} = \begin{vmatrix} 1 & \dots & u_1^{L-2} & \overleftarrow{B}_{L-1}(u_1) & \dots & \overleftarrow{B}_{L-\ell}(u_1) \\ 1 & \dots & u_2^{L-2} & \overleftarrow{B}_{L-1}(u_2) & \dots & \overleftarrow{B}_{L-\ell}(u_2) \\ \vdots & \vdots & \vdots & \vdots & \vdots & \vdots \\ 1 & \dots & u_{L+\ell-1}^{L-2} & \overleftarrow{B}_{L-1}(u_{L+\ell-1}) & \dots & \overleftarrow{B}_{L-\ell}(u_{L+\ell-1}) \end{vmatrix}. \quad (3.10)$$

When there is no enlarging, $\overleftarrow{\mathbb{V}}$ is no longer an operator acting on the Q-functions, but rather the familiar Vandermonde determinant that appears in $\mathfrak{sl}(2)$ SoV orthogonality at leading order. The expression (3.9) highlights that $\det(\mathcal{M}_{L,\ell})$ is a bilinear form in the wavefunctions (products of Q-functions), which is a necessary requirement for any scalar product on the space of states.

Despite the non-democratic nature of the two states in this formula, such expressions are familiar from the separation of variables of higher rank $\mathfrak{sl}(n)$ spin chains [26], whose scalar products also feature Q-functions entering in a non-democratic fashion. The Q-functions for one state enter as simple products, while the others are acted on by certain shift operators. In those setups, this lack of symmetry between the two states can be traced to the fact that one state lives in the fundamental representation, while the other is anti-fundamental, see [27]. In our set-up, the origin of the asymmetry is less clear.

## 3.2 Properties

**Derivation.** There is a lot to unpack in this result, so let us start by addressing how the matrices (3.5) and measures (3.8) were obtained. Similarly to the twist-two case at N²LO, we started constructing the enlarged matrices following the three rules:

   **Rule 1.** All Q-functions are of the form (2.12), with general parameter $\alpha$.

   **Rule 2.** Matrix elements are built only of Q-functions and lower-length Baxter operators.

   **Rule 3.** The matrix elements have no explicit coupling dependence.

For the measure, we followed [36] and the twist-two result (2.43), by also assuming that the perturbative corrections to the measures can be parametrized in the following way

$$\mu_\ell(u) = \mu_\ell^{(0)}(u) \sum_{n=0}^{\infty} \sum_{m=0}^{n} (g\pi)^{2n} a_{n,m} \tanh^{2m}(\pi u). \quad (3.11)$$

By demanding orthogonality among different states at a fixed order in perturbation theory, one can completely fix the unknown coefficients $a_{n,m}$ of the measures (3.11).[12] We then

---

[12]Some coefficients of the measures multiply terms that are zero via lower-order orthogonality. These coefficients, of course, are not fixed. They are completely irrelevant for orthogonality and only affect the relative factor between the norm of the states and the Gaudin norm, which we do not make precise in this work.

verified that these fixed measures yield orthogonality for a plethora of further states (where we avoid problematic cases, see below).[13] In this way, the proposal (3.7) is not derived from some underlying principles, instead at each order in perturbation theory, we can *prove it by exhaustion.* We provide a MATHEMATICA notebook that computes these determinants (3.7) for any two states at any perturbative order before wrapping corrections, there we also present non-trivial examples, such as the $N^5LO$ orthogonality between two twist-five states of spins six and eight.

After considering orthogonality up to $N^5LO$ in the weak-coupling expansion, we started to notice patterns emerging in the coefficients of the measures. Namely, we observed that the measure exponentiates, and that the coefficients of $\tanh(\pi u)$ can be written in the closed form presented in (3.8). To give further evidence that this pattern keeps holding at higher orders in perturbation theory, we then checked that these measures make twist-six states of spins two and four orthogonal at $N^6LO$.[14] This supports the idea that the patterns of the measure persist at higher orders in perturbation theory, allowing us to conjecture the all-loop expression (3.8).

**Lack of Orthogonality for Equal-Spin States.** To make our expressions precise, we must address what we mean by *avoiding problematic cases.* Note that in our main result (3.7) we write a Kronecker delta in the *spins* rather than in the *states.* For twist-two operators, the two conditions are interchangeable, since for each value of spin, there is just a single state. On the other hand, at higher twists, different operators of the same spin start appearing. The major drawback of our proposal (3.7) is that *states of equal twist and equal spin are not orthogonal!* For example, twist-three operators have five states with spin 12; these five states are orthogonal to all other twist-three states, but they are not orthogonal among themselves – this is also exemplified in the attached MATHEMATICA notebook.

This lack of orthogonality among states of the same spin is deeply related with the integrals involving the lower-length Baxter operators, which introduce orderings in the spins of the states (2.39). Such integrals become more and more complicated as we increase the twist, the length of the lower-length Baxter operator, and the perturbative order. This means that we quickly lose the ability to write closed expressions like (2.39), and even in the simplest case of twist-three operators at leading order, we lose track of the analytic spin dependence in the enlarged matrices. Without knowing precisely how the spins of the states affect the enlarged matrices, there is no way to better understand or even solve this problem of a lack of orthogonality for operators with the same values of spin.

**Orthogonality for States of Different Twists.** Notably, provided we avoid states with the same spin, our proposal (3.7) also makes states of *different twists* orthogonal to each other. This is an important property of any putative finite-coupling SoV formalism. The SoV formalism for rational spin chains has the length $L$ fixed. But at finite coupling in $\mathcal{N} = 4$ SYM, the twist is not a quantum number, hence states with different twists will mix. The relevant $\mathfrak{psu}(2, 2|4)$ quantum number is R-charge, which becomes the twist $L$ in the weakly coupled $\mathfrak{sl}(2)$ sector that we consider here. Hence, two states with different R-charge must be orthogonal, and so must be orthogonal in the SoV formalism, which is what we

---

[13]For higher twist at higher loop orders, computing these enlarged determinants becomes numerically challenging, but it is relatively easy to verify the orthogonality evaluations at $N^5LO$ for all states up to twist five and spin ten, which are already 178 different states.

[14]The numerical evaluation becomes heavy at this loop order – orthogonality for operators of twist-six at $N^6LO$ is the determinant of an $11 \times 11$ matrix. Hence, we restricted ourselves to spins two and four for this check.

observe. For example, the leading-order orthogonality of a twist-two state with spin four and a twist-three state of spin six is given by

$$\det(\mathcal{M}[Q_{L=2,S=4}, Q_{L=3,S=6}]) = \begin{vmatrix} 0.548 & 1.014 & -11.226 \\ 3.436 & 6.360 & -70.433 \\ 16.199 & 30.034 & -332.084 \end{vmatrix} = 0\,.$$

Without including wrapping effects, our proposal is far from capturing such finite-coupling mixing results, but it does encode the perturbative version of that (orthogonality of operators with different lengths) in a very simple way. To see this, note that in the lower-length Baxter operator (2.21), we only included the universal term $u^M$ of the transfer matrix of the full Baxter operator (3.1). This is the *only term* of the transfer matrix that matters for the enlarged orthogonality. We could have included the complete transfer matrix of (3.1), but then all lower powers of $u$ can be removed from the enlarged matrices (3.5) via linear transformations of the columns. Therefore, for the purpose of orthogonality, the lower-length Baxter (2.21) with only the universal transfer matrix term and the full Baxter operator (3.1) with all powers of $u$ are equivalent. We opt to write the lower-length Baxter operators with only the universal term to make manifest that *all state-dependent information* of our enlarged orthogonality enters solely through the Q-functions.

When we consider the overlap of states with different twists, the state with the smallest twist will be annihilated by one of the lower-length Baxter operators. This produces entire columns of zeros in the enlarged matrices, making their determinants vanish, and the states trivially orthogonal. This also provides a nice physical interpretation for each of the enlargings we consider: Matrices with no enlarging are the usual orthogonality that makes twist-$L$ states orthogonal among themselves, the first enlarging introduces $B_{L-1}$, making twist-$L$ states orthogonal with twist-$(L-1)$ states, the second enlarging introduces $B_{L-2}$, making them orthogonal with twist-$(L-2)$ states, and so on, until we reach the vacuum state with zero twist.

**Wrapping.** One important aspect of our proposal (3.7) is that we cannot enlarge the matrices forever; the matrix $\mathcal{M}_{L,\ell}$ can only be enlarged *at most L times*. This means that twist-two can be enlarged two times, twist-three enlarged three times, and so on. Since each enlargement corresponds to orthogonality up to a particular order in perturbation theory, an equivalent statement is that twist-two operators can be made orthogonal up to N$^2$LO, twist-three operators up to N$^3$LO, and so on. These are exactly the orders in perturbation theory where wrapping corrections for these operators start to contribute. In other words, the enlarged matrices define perturbative orthogonality up to wrapping order.

**Result.** We can summarize the claim (3.7) and all the properties discussed above as the following statement:

For any two states of twists $L$ and $M$, with $L \geq M$, excluding states of equal twist and equal spin, there is a set of universal measures (3.8) and $L + 1$ enlarged matrices (3.5): $\{\mathcal{M}_{L,0}, \mathcal{M}_{L,1}, \ldots, \mathcal{M}_{L,L}\}$, whose determinants (3.7) vanish at $\{\text{LO}, \text{NLO}, \ldots, \text{N}^L\text{LO}\}$, respectively.

**Finite Coupling Measure.** It is easy to perform the infinite sums appearing in the perturbative measure (3.8) and write them in a closed expression in terms of the 't Hooft

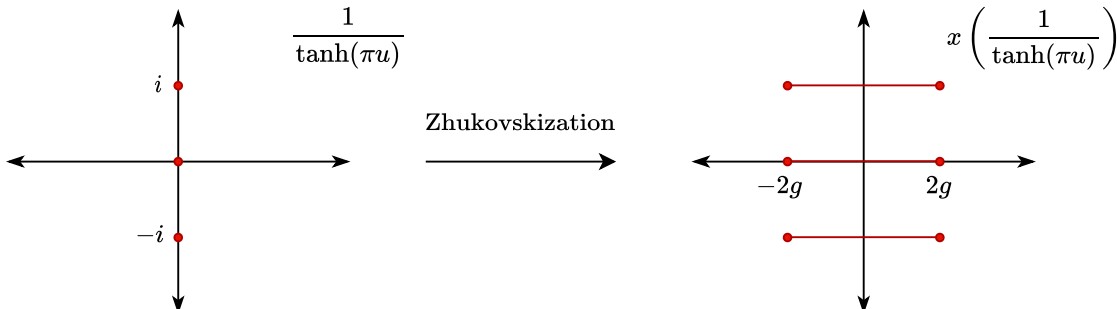

**Figure 1:** Zhukovskization of the measure opens up the $i$-periodic poles into an infinite ladder of cuts between $[-2g, 2g]$.

coupling:

$$\mu_\ell(u) = \frac{\ell(2\pi)^\ell \tanh^{\ell-1}(\pi u)}{\cosh^2(\pi u)} \times \left( \frac{1}{1 + \sqrt{1 + (2\pi g \tanh(\pi u))^2}} \right)^{\ell+1} \times \qquad (3.12)$$

$$\times \exp\left[ g^2 \pi^2 \left( \ell + \frac{2(\ell+4)}{3} \frac{1}{1 + \sqrt{1 + (2\pi g \tanh(\pi u))^2}} - \frac{4(\ell+1)}{3} \frac{1}{\sqrt{1 + (2\pi g \tanh(\pi u))^2}} \right) \right].$$

The first term is the usual tree-level measure, while the other two terms come from summing the two binomials in (3.8). With the help of Zhukovsky variables, we can rewrite the square roots of the above expression in a relatively compact finite-coupling expression

$$\mu_\ell(u) = \frac{\ell}{2\pi \sinh^2(\pi u)} \frac{1}{x(\tau)^{\ell+1}} \exp\left[ g^2 \pi^2 \left( \ell + \frac{4+\ell}{3} \frac{\tau}{x(\tau)} + \frac{4(\ell+1)}{3} \frac{1}{1 - 2x(\tau)/\tau} \right) \right], \quad (3.13)$$

where $\tau(u) = 1/(\pi \tanh(\pi u))$. This Zhukovsky map of the measure is similar to the Zhukovskization procedure considered in [35], its effect is to promote some of the $i$-periodic poles of the measure to $i$-periodic cuts, as depicted in Figure 1.

Spoiling the beautiful result in terms of Zhukovsky variables (3.13) is the exponential, which depends explicitly on $\tau(u)$ and the coupling. Perhaps, by correctly incorporating wrapping effects into the SoV framework, one could generate terms that combine with this exponential and result in an expression for the measure where the coupling dependence enters only via the Zhukovsky variables. Obtaining such finite-coupling measures is the holy grail of the SoV formalism in $\mathcal{N} = 4$ super Yang–Mills, and is what is missing to elevate it to the same standards as the Quantum Spectral Curve.

Unfortunately, our proposal (3.7) is not there yet. It does not make states with the same spin orthogonal, and its leading-order relation with the Gaudin norm changes with each successive enlargement (see Appendix B). To resolve these problems, we would need to have complete analytic control over these enlarged orthogonality expressions at N²LO and beyond, which we are currently lacking. In the next section, we provide an *alternative* enlarged orthogonality formulation, where we do have such analytic control, and we comment on how some of these problems can be resolved.

# 4  Orthogonality With Residues

## 4.1  SoV with One Q-Function

To develop this alternative formulation, we go back to twist-two operators. Our starting point will be equation (2.20), but we will make it more democratic in the Q-functions by symmetrizing it in the states:

$$\Delta I_0 \langle \mathbb{Q}_S \mathbb{Q}_J \rangle_\mu + g^4 \frac{\Delta q_2^+}{4} \langle B_0[\mathbb{Q}_S, \mathbb{Q}_J] \rangle_\mu + \texttt{res}_\mu = 0\,, \tag{4.1}$$

with $B_M[F,G] \equiv (B_M \cdot F)G + (B_M \cdot G)F$ being the symmetric action of lower-length Baxter operators (2.21), and the Q-functions are given by (2.12) with $\alpha = 1/2$ as dressing parameters. The residues that are picked by the contour shifts are given by

$$\texttt{res}_\mu = \operatorname*{Res}_{u=i/2} \left[ \mathbb{Q}_S(u) \left( (x^-)^2 - \frac{g^4}{4} \Big( q_2^+(S) + q_2^+(J) \Big) \right) \mathbb{Q}_J(u-i)\,\mu(u) \right]$$
$$+ \operatorname*{Res}_{u=-i/2} \left[ \mathbb{Q}_S(u) \left( (x^+)^2 - \frac{g^4}{4} \Big( q_2^+(S) + q_2^+(J) \Big) \right) \mathbb{Q}_J(u+i)\,\mu(u) \right]\,. \tag{4.2}$$

As previously recalled in Section 2, the guiding idea of the Separation of Variables formalism is to find as many measures with vanishing residues as integrals of motion and conserved charges appear in the equation.

For any given measure, it is possible to compute the residue (4.2) in terms of the conserved charges $q_n^\pm(S)$ and $q_n^\pm(J)$, *without* having to specify their on-shell values in terms of Bethe roots. Since these charges are independent, by demanding that the coefficient of each of them is zero, we can find measures with vanishing residues in a fully off-shell way. We explain this procedure further in Appendix C, and it trivializes the searching for SoV measures. For example, in the twist-two case (4.2), we find that up to N²LO, there exists a single measure with vanishing residues, given by

$$\rho_1(u) = \frac{\pi/2}{\cosh^2(\pi u)} \Big( 1 - g^2 \pi^2 (1 - 3\tanh^2(\pi u)) + \frac{2}{3} g^4 \pi^4 (2 - 15\tanh^2(\pi u) + 15\tanh^4(\pi u)) \Big). \tag{4.3}$$

Since the conserved charge $q_2^+$ appears only at N²LO, up to NLO there is a single integral of motion $I_0$, and therefore a single measure is sufficient to define orthogonality. In other words, the measure (4.3) defines next-to-leading order orthogonality[15]

$$\mathbb{M}_{S,J}^{(1)} = \left( \langle \mathbb{Q}_S \mathbb{Q}_J \rangle_{\nu_1} \right)\,, \quad \det \left( \mathbb{M}_{S,J}^{(1)} \right) \propto \delta_{S,J} + \mathcal{O}(g^4)\,. \tag{4.4}$$

Another way of writing this measure is via the *Zhukovskization* procedure [35]. By convoluting the leading order term with the Zhukovsky variable, we can recover all its perturbative expansion via the simple contour integral

$$\rho_1(u) \equiv \oint \frac{dv}{2\pi i} \frac{\pi/2}{\cosh^2(\pi(u-v))} \frac{1}{x(v)}\,, \tag{4.5}$$

which is a similar way of promoting these measures beyond perturbation theory, by opening the poles into cuts as depicted in Figure 1. Although exciting, we will not consider finite-coupling aspects any further, since the orthogonality we develop below, so far, is only applicable at N²LO in the context of twist-two operators.

---

[15]The first perturbative order to (4.3) is exactly the same as the one proposed in [36]. The second perturbative order of the measure (4.3) is equal to the first line of equation (14) in [36], which is the state-independent part of the measure considered there.

Two measures with vanishing residues are needed to define N²LO orthogonality, but only one measure that enjoys such a property exists in the setup (4.1). As explained before, there are many directions one can pursue in the SoV framework to address this problem. In Section 3, we took a more numerical approach, bypassing all residue computations, and searching directly for orthogonal measures. Here, we follow a more analytical approach by focusing on other ways that the residues can yield orthogonality for the Q-functions.

Our explorations will rely on the following rather simple observation: Since the goal is to have as many measures as integrals of motion and conserved charges appear in the equation, the residues do not necessarily need to be zero, but themselves can also given by a combination of these (differences of) integrals of motion and conserved charges. More explicitly, in the equation (4.1), we can extend our search from measures with vanishing residues to measures whose residues are given by the difference $\Delta q_2^+$ of the conserved charge $q_2^+$, multiplied by integrals with state-independent measures.

## 4.2  SoV with Two Q-Functions

Even though we enlarged our search for measures, it turns out that no measure generates residues proportional to *only* $\Delta q_2^+$, other conserved charges $q_n^+$ also appear. But there is a way out of this as well. In all higher-rank integrable models, more than one type of Q-functions that solve the Baxter equation appear in the SoV orthogonality relation [26–29]. However, in rank-one sectors such as $\mathfrak{sl}(2)$, the second solution to the Baxter equation often plays no role. As we explain below, by including this second Q-function, we will be able to consider another measure $\rho_2(u)$, whose residues are proportional only to the conserved charge $\Delta q_2^+$. This will allow us to close the linear system (4.1) and define a new SoV-type expression for N²LO orthogonality for twist-two operators.

One main difference between the two Q-functions lies in their different large-$u$ asymptotic behavior: While the Q-function we have considered so far has asymptotics $Q_S^{(1)}(u) \sim u^S$, the second solution behaves as $Q_S^{(2)}(u) \sim u^{-S-1}$. Both objects can be equally easily computed via the perturbative Quantum Spectral Curve [45]. For example, the two Q-functions for the twist-two operator of spin two at leading order are

$$Q_{S=2}^{(1)}(u) = \frac{1}{4} - 3u^2 \,, \tag{4.6}$$

$$Q_{S=2}^{(2)}(u) = -3iu + \left(\frac{1}{4} - 3u^2\right)\psi_1\left(\frac{1}{2} + iu\right) . \tag{4.7}$$

Both Q-functions satisfy the same Baxter equation (2.1), and so the Baxter equation for any solution $Q(u)$ can be written as the following trivial determinant

$$\begin{vmatrix} Q(u+i) & Q(u) & Q(u-i) \\ Q^{(1)}(u+i) & Q^{(1)}(u) & Q^{(1)}(u-i) \\ Q^{(2)}(u+i) & Q^{(2)}(u) & Q^{(2)}(u-i) \end{vmatrix} = 0\,. \tag{4.8}$$

Expanding this determinant and comparing with the usual Baxter equation (2.1) then fixes the Wronskian at leading-order to be

$$Q_S^{(1)}\left(u + \frac{i}{2}\right) Q_S^{(2)}\left(u - \frac{i}{2}\right) - Q_S^{(1)}\left(u - \frac{i}{2}\right) Q_S^{(2)}\left(u + \frac{i}{2}\right) = \frac{1}{u^2}\,. \tag{4.9}$$

In this section, we will use a different overall normalization for the Q-functions (2.12) than we use elsewhere in the paper. This normalization makes the residue computations

simpler, as explained in Appendix C, and is achieved by

$$\hat{Q}_S^{(1)}(u) = \frac{1}{\mathcal{P}_S(i/2)} Q_S^{(1)}(u). \tag{4.10}$$

Note that choosing this normalization automatically adds the factor $\mathcal{P}_S(i/2)$ to the second Q-function, since their product must result in the Wronskian relation (4.9),

$$\hat{Q}_S^{(2)}(u) = \mathcal{P}_S(i/2) Q_S^{(2)}(u). \tag{4.11}$$

This normalization is precisely the one that is presented in (4.6) and (4.7) for the spin-two operator.

Since $Q_S^{(2)}$ is a solution to the same Baxter equation as $Q_S^{(1)}$, it also satisfies the same equation as (4.1), explicitly:

$$\Delta I_0 \langle \mathbb{Q}_S^{(2)} \mathbb{Q}_J^{(2)} \rangle_\mu + g^4 \frac{\Delta q_2^+}{4} \langle B_0[\mathbb{Q}_S^{(2)}, \mathbb{Q}_J^{(2)}] \rangle_\mu + \texttt{res}_\mu = 0. \tag{4.12}$$

However, the different asymptotics of this second Q-function allows us to search for periodic measures that do not decay exponentially at infinity. In fact, we find that the measure

$$\rho_2(u) = \pi \tanh(\pi u) \tag{4.13}$$

in the equation (4.12) for $Q_S^{(2)}$, generates residues that are given by

$$\texttt{res}_{\rho_2} = \frac{\Delta q_2^+}{4} \times 1 = \frac{\Delta q_2^+}{4} \times \mathcal{P}_S\left(\frac{i}{2}\right) \mathcal{P}_J\left(\frac{i}{2}\right), \tag{4.14}$$

where we explicitly wrote the number one in terms of the Q-functions, to make manifest how changing the overall normalization of the Q-functions changes our expressions.

Thus, using the two Q-functions $Q_S^{(1)}$ and $Q_S^{(2)}$, and the two measures (4.3) and (4.13), we can write the following linear system, where the r.h.s. emphasizes to which perturbative order each equation holds:

$$\begin{cases} \Delta I_0 \times \langle \hat{\mathbb{Q}}_S^{(1)} \hat{\mathbb{Q}}_J^{(1)} \rangle_{\rho_1} + \dfrac{\Delta q_2^+}{4} \times g^4 \langle B_0[\hat{\mathbb{Q}}_S^{(1)}, \hat{\mathbb{Q}}_J^{(1)}] \rangle_{\rho_1} &= \mathcal{O}(g^6) \\ \Delta I_0 \times \langle \hat{\mathbb{Q}}_S^{(2)} \hat{\mathbb{Q}}_J^{(2)} \rangle_{\rho_2} + \dfrac{\Delta q_2^+}{4} \times 1 &= \mathcal{O}(g^2) \end{cases}, \tag{4.15}$$

which we can transform into the matrix equation

$$\mathbb{M}_{S,J}^{(2)} \begin{pmatrix} \Delta I_0 \\ \frac{\Delta q_2^+}{4} \end{pmatrix} = \begin{pmatrix} \mathcal{O}(g^6) \\ \mathcal{O}(g^2) \end{pmatrix}, \qquad \mathbb{M}_{S,J}^{(2)} = \begin{pmatrix} \langle \hat{\mathbb{Q}}_S^{(1)} \hat{\mathbb{Q}}_J^{(1)} \rangle_{\rho_1} & g^4 \langle B_0[\hat{\mathbb{Q}}_S^{(1)}, \hat{\mathbb{Q}}_J^{(1)}] \rangle_{\rho_1} \\ \langle \hat{\mathbb{Q}}_S^{(2)} \hat{\mathbb{Q}}_J^{(2)} \rangle_{\rho_2} & 1 \end{pmatrix}. \tag{4.16}$$

While the equation as written looks normalization dependent, it is easy to restore the normalization invariance by rewriting the number one in the lower-right in terms of Q-functions as in (4.14) and noticing that the overall $\mathcal{P}_S(i/2)\mathcal{P}_J(i/2)$ factors out.

One might be worried that perturbative corrections for the measure $\rho_2(u)$, residues, and Q-functions $Q_S^{(2)}$ need to be considered in the bottom row. But, as we will explain now, such corrections do not matter for N²LO orthogonality.

If the two states are different, the first matrix element is already order $\mathcal{O}(g^4)$, as shown in (4.4). Thus, the entire first row of the matrix is of order $\mathcal{O}(g^4)$, which washes away any perturbative corrections that the bottom row might have when computing the determinant.

The matrix (4.16) is another SoV-type representation for N²LO orthogonality for twist-two operators[16]

$$\det\left(\mathbb{M}_{S,J}^{(2)}\right) \propto \delta_{S,J} + \mathcal{O}(g^6).$$ (4.17)

If the states are equal, the first matrix element is order $\mathcal{O}(g^0)$, and therefore corrections to $Q_S^{(2)}$, measures and residues must be taken into account to make precise the connection between the Gaudin norm and the SoV proposal (4.17). Nonetheless, the fact that the lower-right matrix element is $1 + \mathcal{O}(g^2)$ guarantees that, despite the perturbative corrections, the enlarged matrix $\mathbb{M}_{S,J}^{(2)}$ and the original-size matrix $\mathbb{M}_{S,J}^{(1)}$ will have the *same* normalization factor with the Gaudin norm at leading order.[17]

The explicit coupling dependence on the matrix elements makes clear the difference between the two SoV type expressions for twist-2 proposed in this work: (2.42) and (4.17). Before we had no terms with explicit coupling dependence, which led to an obscure normalization with the Gaudin norm. Here, the residues enter the enlarged matrix with explicit coupling dependence, resulting in a more transparent connection with the Gaudin norm at leading order.

The drawback of this proposal is that is very difficult to search for measures whose residues are proportional to conserved charges, especially for the case involving $Q_S^{(2)}$, whose residues we do not know how to write in a off-shell way like $Q_S^{(1)}$. In practice, what we do is to compute the residues on-shell for many different states, and then try to recast them as a state-independent combination of conserved charges.

This proportionality is only unambiguous if we have a precise map between the conserved charges and the Q-functions. Otherwise, any state-dependent change in the normalization of the Q-functions would change the proportionality with the conserved charges. Such maps exist for $Q_S^{(1)}$, for any coupling and any twist, see (A.8). But for $Q_S^{(2)}$ we were only able to find the relation of this Q-function with $q_2^+$ for the case of twist-2 operators at leading order. Without such maps for higher twists and higher orders in perturbation theory, the concept of proportionality becomes ill-defined, and we were not able to push these ideas for more complicated cases than twist-2 operators at N²LO. The proposal (4.17) must be understood as a *proof of concept* to call attention to the fact that SoV explorations can include more than one type of Q-functions (even in rank-1 sectors) and show that it is possible to include non-vanishing residues in the SoV framework.

# 5 Discussion

In this work, we proposed a Separation of Variables type expression for orthogonality in the $\mathfrak{sl}(2)$ sector of $\mathcal{N} = 4$ super Yang–Mills theory, valid to all orders in the weak-coupling expansion prior to wrapping corrections. More precisely, we defined a set of ever-growing matrices (3.5), together with a set of finite-coupling measures (3.12), that make two states of non-equal spins orthogonal up to the perturbative order at which wrapping corrections start contributing.

Besides not capturing orthogonality for states of equal spin, each successive enlarging changes the proportionality factor with the Gaudin norm, which makes it difficult to precisely establish the connection between the SoV determinants proposed here and the two-point functions of operators.

---

[16]To be completely sure, we verified the orthogonality relation (4.17) for states up to spin $S = 60$.

[17]Better understanding these corrections could shine light on the mysterious reciprocity-like normalization factor $\mathbb{J}$ between SoV and Gaudin norm observed in [36].

We attempted to resolve these issues by generalizing the properties of the residues and considering both type of Q-functions for the $\mathfrak{sl}(2)$ sector, the latter of which are already a necessary ingredient in the SoV expressions of other models [47–52]. The outcome was another SoV-type expression (4.17), which does not change the normalization factors with the Gaudin norm. So far, we could not generalize that orthogonality expression to operators of arbitrary twist at higher orders in perturbation theory, which would be required to check whether the orthogonality among operators of equal spin is resolved as well.

We believe that the general lessons drawn from the two SoV-inspired approaches we developed in this work could serve as guidelines for future explorations involving the Separation of Variables formalism in $\mathcal{N} = 4$ super Yang–Mills theory as well as other integrable models. Let us review the salient points:

**1. Dressings.**  In perturbation theory, the transfer matrix has non-polynomial parts, whose coefficients are controlled by the dressing $e^{\alpha \cdot \sigma(u)}$ of the Q-functions. There is no apparent conceptual problem in considering the resulting extra poles in the SoV formalism, but to our knowledge, thus far no results have accounted for such poles. Our proposal offers a simple way of removing these terms: *Make the non-polynomial part of the transfer matrix universal.* Which is precisely what the QSC-inspired dressing (2.10) does in the asymptotic Baxter equation. Identifying whether these dressed Q-functions correspond to natural objects in the context of full QSC – which is important in order to understand how to incorporate wrapping effects – remains an outstanding question.

**2. Q-functions.**  The second orthogonality formulation we proposed implies that even in rank-one sectors such as $\mathfrak{sl}(2)$, all different types of Q-functions that solve the Baxter equation should be considered, as is the case for higher-rank integrable models [25, 26]. Moreover, in both orthogonality formulations, we see a new ingredient appearing in the SoV framework, namely lower-length Baxter operators, which appear at each order in perturbation theory. This suggests that the perturbative SoV framework for $\mathcal{N} = 4$ super Yang–Mills theory, even at rank one, should depend on *all Q-functions, as well as lower-length Baxter operators acting on them, with appropriate transcendentality.*

**3. Residues.**  To find Functional SoV orthogonality relations, one must obtain as many measures as integrals of motion and conserved charges appear in the equation. In order to not introduce extra terms in this counting, one usually searches for measures that have vanishing residues. However, there are some previously overlooked measures which do have residues, but still do not change this counting. Namely, *measures might have non-zero residues, as long as they are proportional to differences of integrals of motions and conserved charges, multiplied by bilinear integrals of Q-functions with modified (state-independent) measures.*

It would be nice to revisit known integrable models that have elusive SoV representations, for example, the supersymmetric $\mathfrak{su}(m|n)$ spin chain, or the $\mathfrak{su}(2)$ XXZ spin chain, which so far admit separated wave functions [24, 53], but whose correlation functions remain out of reach. Perhaps in these cases, one also needs to relax conditions on the measures and residues compared to the rational $\mathfrak{su}(n)$ set-up.

Although orthogonality seems to be the correct guideline to search for SoV representations for two-point functions, there is no guarantee that the measures we obtained here will be useful for three-point functions. For example, in the $\mathfrak{su}(2)$ sector, the measures must be slightly modified when going from two-point to three-point functions [36]. The SoV-inspired proposal (3.7) can account for this difference in a neat way: Due to the asymmetry in $S$ and

$J$, the determinant expressions

$$\det \mathcal{M}_{L,\ell}[Q_S, Q_J]\,, \qquad \det \mathcal{M}_{L,\ell}[Q_J, Q_S]\,, \qquad \sqrt{\det \mathcal{M}_{L,\ell}[Q_S, Q_J] \det \mathcal{M}_{L,\ell}[Q_S, Q_J]} \quad (5.1)$$

are all different, and while it is natural that their symmetric combination could be used to compute two-point functions, the other non-symmetric determinants could compute the overlaps with the vacuum and three-point functions.

One might be worried that the unstable relative factor between the SoV expressions and the Gaudin norm spoils such explorations, but one hope is that these factors become simple when considering three-point functions. This is exactly what was observed for the $\mathfrak{sl}(2)$ and $\mathfrak{su}(2)$ sectors: The normalization factors between SoV expressions for the norm $\mathbb{B}$ and the SoV overlaps with the vacuum (the so-called $\mathcal{A}$ of the integrable hexagons framework [4]) were built from complicated combinations of Bethe roots. On the contrary, the normalization factor of the ratio of these two quantities, which is precisely what evaluates three-point functions, was given by a simple factorial [36]. Perhaps more about the relation between these objects can be gained by studying the orbifolded or twisted setups recently analyzed using hexagons [9].

The all-loop expressions we proposed here could open a new way for strong-coupling explorations in the Separation of Variables formalism. For small operators, our expressions break down due to wrapping corrections, and we have nothing to say about strong coupling. But for parametrically large operators, our expressions should remain valid, and perhaps there could be some interesting semi-classical limits emerging in the SoV formalism, like the ones observed in [31, 54–57].

Another interesting possibility is to consider an analytic continuation in the spin $S$. The Baxter equation for twist three at complex spin was recently extensively analyzed in [58]. As was argued in [59], one cannot simply upgrade the SoV expression for the three-point function built from $Q_S^{(1)}$ by promoting it to its entire analytic continuation [60], built from both $Q^{(1)}$ and $Q^{(2)}$ with $i$-periodic coefficients. This may be related to the fact that in the QSC, there are two distinct objects which become $Q^{(1)}$ for integer $S$ at tree level, but which differ as soon as $S$ is complex.[18] It would be nice to understand how these analytic continuations fit in the SoV framework. Perhaps the relevant objects for a complex $S$ SoV expression are neither $Q^{(1)}$ nor $Q^{(2)}$, but precise linear combinations of these two Q-functions that have yet to be found.

Another limit that would be nice to consider is the large-spin limit. This limit is another mechanism that could suppress the wrapping corrections, and still be valid for small-twist operators. Differently from the integrable hexagons formalism [4], where the spin controls the number of partitions one needs to sum over, in the SoV expressions the spin only controls the degree of the polynomial part of the Q-functions. This makes SoV-type expressions amenable to both numerical as well as analytical explorations at large spin. Better understanding this limit could shine further light on the dualities between Wilson loops and correlation functions that are present in $\mathcal{N} = 4$ super Yang–Mills [61–64].

We hope that our proposal, or at least the general lessons we are suggesting, will help to improve the Separation of Variables framework in $\mathcal{N} = 4$ super Yang–Mills and other integrable models, and bring us one step closer to the same level of computational efficiency as the Quantum Spectral Curve.

---

[18]These are $Q_{12|12}$ discussed in the main text, and another object denoted $\mu_{12}^+$.

# Acknowledgments

We are grateful to B. Basso, J. Brödel, S. Ekhammar, N. Gromov, A. Homrich, E. Im, G. Lefundes, A. Pribytok, N. Primi, D. Serban, R. Tateo, P. Vieira, and D. Volin for useful discussions. We are especially grateful to A. Homrich and P. Vieira for numerous suggestions, and collaboration on several topics discussed here. The work of T. B., C. B., D. L. and P. R. was funded by the Deutsche Forschungsgemeinschaft (DFG, German Research Foundation) Grant No. 460391856. T. B., C. B., D. L. and P. R. acknowledge support from DESY (Hamburg, Germany), a member of the Helmholtz Association HGF, and by the Deutsche Forschungsgemeinschaft (DFG, German Research Foundation) under Germany's Excellence Strategy – EXC 2121 "Quantum Universe" – 390833306. D. L. further thanks University of Turin and ETH Zürich for the generous and repeated hospitality, Antonio Antunes for his invaluable insights into the history of integrability and Lucia Bianco for her unique and unwavering support. A. C. participates to the project HORIZON-MSCA-2023-SE-01-101182937-HeI, and acknowledges support from this action and from the INFN SFT specific initiative. He also thanks DESY for the warm hospitality during two extended visits.

# A    Bethe equations

The Bethe roots $v_n$, $n = 1, \ldots, S$, are solution to the asymptotic Bethe ansatz (ABA) equations [10, 65]

$$\left(\frac{x_k^+}{x_k^-}\right)^L = \prod_{j \neq k}^S \frac{x_k^- - x_j^+}{x_k^+ - x_j^-} \frac{1 - g^2/x_k^+ x_j^-}{1 - g^2/x_k^- x_j^+} e^{2i\,\theta(v_k, v_j)} \,. \tag{A.1}$$

In addition to the ABA equations, one also needs to impose cyclicity of the trace in the form of the zero-momentum condition

$$\prod_{k=1}^S \frac{x_k^+}{x_k^-} = 1 \,, \tag{A.2}$$

where the Zhukovsky variables are

$$x_k^\pm = x\left(v_k \pm \frac{i}{2}\right) \quad \text{with} \quad x(u) = \frac{u + \sqrt{u^2 - 4g^2}}{2} \,. \tag{A.3}$$

The object $\theta(u, v)$ is the so-called dressing phase [65–71]. In weak-coupling perturbation theory, the phase factor starts to contribute only at order $\mathcal{O}(g^6)$, and is built out of several infinite sums. At any finite order $\mathcal{O}(g^{2\Lambda})$, these sums truncate, and the dressing phase can be written as

$$\theta(u, v) = \sum_{r=1}^\Lambda \sum_{s=1}^\Lambda \sum_{n=0}^{\Lambda-r-s} g^{2(r+s+n)} \beta_{r,s,n} t_r(u) t_s(v) \,, \tag{A.4}$$

$$\beta_{r,s,n} = 2(-1)^n \frac{\sin(\frac{\pi}{2}(r-s))\zeta_{2n+r+s}\Gamma(2n+r+s)\Gamma(2n+r+s+1)}{\Gamma(n+1)\Gamma(n+r+1)\Gamma(n+s+1)\Gamma(n+r+s+1)} \,, \tag{A.5}$$

$$t_r(u) = \left(\frac{1}{x^+(u)}\right)^r - \left(\frac{1}{x^-(u)}\right)^r \,. \tag{A.6}$$

Using an extension of the following MATHEMATICA code presented in [72],

```
ns[L_,S_] = Select[Subsets[Range[-(L+S-3)/2, (L+S+3)/2], {S}],
    Mod[Total[#], L] == 0 &];
BAE[L_, S_, n_] := Table[2 L ArcTan[2u[j]] + 2Sum[If[j==k,0,
```

```
        ArcTan[u[j] - u[k]]],{k,1,S}] - 2 Pi ns[L,s][[n,j]],{j,1,S}]
  U[L_,S_] := Table[#[[2]] &/@ FindRoot[BAE[L,S,n], Table[{u[j], j-S/2},
      {j, S}], WorkingPrecision -> 150], {n,Length@ns[L, S]}]
```

it is easy to enumerate the $\mathfrak{sl}(2)$ states for any twist $L$ and spin $S$, and then compute their corresponding Bethe roots $v_n$ at leading order with arbitrary numerical precision. After finding the leading-order solutions `U[L,S]` for the Bethe equations (A.1), it is easy to compute their loop corrections by simply linearizing the Bethe equations around these seed values.

In this way one can evaluate the Bethe roots at any order in perturbation theory with arbitrary numerical precision. It is then trivial to assemble them to write the polynomial part of the Q-functions $\mathcal{P}(u) = \prod_{k=1}^{S}(u - v_k)$, and to write the conserved charges that appear in the dressing (2.13), via

$$q_n^{\pm}[Q_S] = \sum_{k=1}^{S} \left( \left(\frac{i}{x_k^+}\right)^n \pm \left(\frac{-i}{x_k^-}\right)^n \right) . \tag{A.7}$$

Since the conserved charges are simple combinations of the Bethe roots, it is also possible to write them as simple contour integrals of the polynomial Q-functions

$$q_n^+ = \oint \frac{\mathrm{d}x}{4\pi i g^{n-1}} (-1)^n \left(1 - \frac{g^2}{x^2}\right) \left(\frac{x^n}{g^n} - \frac{g^n}{x^n}\right) \left(i^n \frac{\mathcal{P}'(u + \frac{i}{2})}{\mathcal{P}(u + \frac{i}{2})} + (-i)^n \frac{\mathcal{P}'(u - \frac{i}{2})}{\mathcal{P}(u - \frac{i}{2})}\right) ,$$

$$q_n^- = \oint \frac{\mathrm{d}x}{4\pi i g^{n-1}} (-1)^n \left(1 - \frac{g^2}{x^2}\right) \left(\frac{x^n}{g^n} - \frac{g^n}{x^n}\right) \left(i^{n-1} \frac{\mathcal{P}'(u + \frac{i}{2})}{\mathcal{P}(u + \frac{i}{2})} + (-i)^{n-1} \frac{\mathcal{P}'(u - \frac{i}{2})}{\mathcal{P}(u - \frac{i}{2})}\right) , \tag{A.8}$$

where the contour is the counter-clockwise unit circle in the $x$-plane. The equivalence with (A.7) can be easily checked by computing the integrals by residues.

# B    Gaudin Norm

The Gaudin norm at leading order for a operator of twist-$L$ and spin $S$ is given by

$$\mathbb{B}_{L,S} = \frac{\det(\partial_{v_i}\phi_j)}{\prod_{i \neq j} h(v_i, v_j)} \tag{B.1}$$

where the derivatives act on the Bethe equations via

$$e^{i\phi_j} = e^{ip(v_j)L} \prod_{k \neq j} S(v_j, v_k), \tag{B.2}$$

and the momenta and S-matrices at leading order are given by

$$e^{ip(u)} = \frac{u + \frac{i}{2}}{u + \frac{i}{2}}, \quad S(u,v) = \frac{h(u,v)}{h(v,u)} \quad \text{and} \quad h(u,v) = \frac{u - v}{u - v + i} . \tag{B.3}$$

The relation between the enlarged orthogonality proposed in (3.7) and the Gaudin norm at leading order for the first two enlargings is

$$\mathcal{D}_{L,0}[Q_S, Q_J] = \delta_{S,J} \times \frac{\Gamma(L)}{\Gamma(2S + L)} \left(Q_S\left(\frac{i}{2}\right) Q_S\left(-\frac{i}{2}\right)\right)^{-L} \times \mathbb{B}_L(S),$$

$$\mathcal{D}_{L,1}[Q_S, Q_J] = \delta_{S,J} \times \frac{\Gamma(L)}{\Gamma(2S + L)} \left(Q_S\left(\frac{i}{2}\right) Q_S\left(-\frac{i}{2}\right)\right)^{-(L+1)} \times \frac{\mathbb{B}_L(S)}{q_1^+(S)} . \tag{B.4}$$

For bigger enlargings, we can easily evaluate the proportionality constant between these two quantities, see (2.40), but we were not able to write it in a closed form in terms of the charges like (B.4).

# C Evaluating Residues

There are two ways to compute the residues that are picked in the contour manipulations of the initial equation (2.3). The first option is to plug the on-shell values of the Q-functions (with all Bethe roots and charges numerically computed), and then evaluate the residue for many different states to check if it is zero, or if it can be written as some combination of charges.

In this appendix we develop an alternative way, where we do not use the explicit values of the Bethe roots and conserved charges for the Q-functions. In this off-shell setup, we compute the residues as analytic functions of the conserved charges. This trivializes the process of finding measures with vanishing residues, and makes it easier to find measures with residues that are proportional to conserved charges. We exemplify this process by considering N²LO orthogonality for twist-two operators.

The residues that are picked in the contour manipulations that transform the initial equation (2.3) to the final equation (2.20) can be written explicitly as

$$\mathtt{res}_\mu = \operatorname*{Res}_{u=i/2} \left[ \mathbb{Q}_S(u) \left( (x^-)^2 - \frac{g^4}{4} \big( q_2^+(S) + q_2^+(J) \big) \right) \mathbb{Q}_J(u-i)\mu(u) \right] + \tag{C.1}$$

$$\operatorname*{Res}_{u=-i/2} \left[ \mathbb{Q}_S(u) \left( (x^+)^2 - \frac{g^4}{4} \big( q_2^+(S) + q_2^+(J) \big) \right) \mathbb{Q}_J(u+i)\mu(u) \right] + (S \leftrightarrow J) \,.$$

Computing this residue for arbitrary measures can be done in three simple steps.

**Step 1.** Consider an ansatz for the measure. It can either be in terms of hyperbolic functions like (2.41)

$$\mu(u) = \frac{\pi/2}{\cosh^2(\pi u)} (1 + g^2(a_0 + a_1 \tanh^2(\pi u)) + \mathcal{O}(g^4)) \tag{C.2}$$

Or, more generally, it can also be the Laurent series of the measure around the poles $u = \pm i/2$, which is the only relevant part to compute the residues

$$\mu(u) = \sum_{n=-\Lambda_0}^{\Lambda_0} \left( \frac{a_n}{\left(u + \frac{i}{2}\right)^n} + \frac{a_n}{\left(u - \frac{i}{2}\right)^n} \right) + g^2 \sum_{n=-\Lambda_1}^{\Lambda_1} \left( \frac{b_n}{\left(u + \frac{i}{2}\right)^n} + \frac{b_n}{\left(u - \frac{i}{2}\right)^n} \right) + \mathcal{O}(g^4) \tag{C.3}$$

where $\Lambda_i$ are cutoffs on the expansion whose values we explain below.

**Step 2.** Plug the Laurent series (C.3) and the Q-functions (2.12) in the residue expression (C.1). This can be done off-shell, namely we do not need to specify the polynomial part of the Q-functions, and neither the values of the charges appearing in the dressing $e^{\alpha \cdot \sigma(u)}$. At N²LO, it is sufficient to write:

$$\mathbb{Q}_S(u) = \mathcal{P}_S(u) \left( 1 + g^2 q_1^+ \psi_0^+ - \frac{g^4}{2} \left( q_1^+ \psi_2^+ + q_2^- \psi_1^- \right) \right) + \mathcal{O}(g^6) \,. \tag{C.4}$$

Since $\mathcal{P}_S(u)$ is regular at $u = \pm i/2$, all poles and zeros of the Q-functions at $u = \pm i/2$ are implemented via the Laurent series of the measure and the off-shell Q-functions, so one can just evaluate the residues (C.1) explicitly.

For example, using the measure (C.2) and the off-shell Q-functions (C.4), we can write the residue (C.1) at NLO as

$$\mathtt{res}_\mu = 0 + i g^2 \mathcal{P}_S \left( \tfrac{i}{2} \right) \mathcal{P}_J \left( \tfrac{i}{2} \right) (q_1^+(S) - q_1^+(J)) +$$

$$+ 2 g^2 (2 - a_1) \left( \mathcal{P}_S \left( \tfrac{i}{2} \right) \mathcal{P}_J' \left( \tfrac{i}{2} \right) - \mathcal{P}_J \left( \tfrac{i}{2} \right) \mathcal{P}_S' \left( \tfrac{i}{2} \right) \right) + \mathcal{O}(g^4) \,. \tag{C.5}$$

**Step 3.** At this stage, the evaluated residues depend on the conserved charges $q_i^\pm$, the coefficients of the measure, and the polynomial $\mathcal{P}_S(u)$ and its derivatives evaluated at $u = \pm i/2$. To make this expression useful, we can use the fact that these polynomial (and their derivatives) at specific values of the spectral parameter are themselves written in terms of conserved charges. For example, fixing the normalization of the Q-functions so that $\mathcal{P}_S(i/2) = 1$, we then have

$$\mathcal{P}_S'\left(\tfrac{i}{2}\right) = \frac{q_1^+}{2i} + g^2 \frac{q_3^+}{2i} + g^4 \frac{q_5^+}{2i} + \mathcal{O}(g^6) \tag{C.6}$$

$$\mathcal{P}_S''\left(\tfrac{i}{2}\right) = \left(\frac{q_2^+}{2} - \frac{(q_1^+)^2}{4}\right) + \frac{g^2}{2}\left(q_4^+ - q_1^+ q_3^+\right) + \frac{g^4}{2}\left(3q_6^+ - q_1^+ q_5^+ - \frac{(q_3^+)^2}{2}\right) + \mathcal{O}(g^6) \tag{C.7}$$

The outcome of implementing these three steps is that for any given cut-offs $\Lambda_i$ (or combination of hyperbolic functions) we can compute the residue (C.1) as a function of the conserved charges of the Q-functions and the coefficients in the measure ansatz. In the case of (C.5), the residue becomes

$$\texttt{res}_\mu = 0 - ig^2(a_1 - 3)(q_1^+(S) - q_1^+(J)) + \mathcal{O}(g^4). \tag{C.8}$$

This in turns fixes the measure at one-loop order to $a_1 = 3$. We conclude that for the dressing parameter $\alpha = 1/2$, at NLO there is a unique measure that has vanishing residues.[19] This statement remains true at the next order, and the resulting unique N²LO measure is (4.3). The same analysis can be done for arbitrary values of $\alpha$, and at any loop order. By doing precisely such computations, we obtain that no two measures with vanishing residues exist for twist-two operators.

In principle, one could try to search for alternative measures by considering an ansatz with arbitrary large values of the cut-offs $\Lambda_i$ (or arbitrarily large powers of $\tanh(\pi u)$ in the ansatz (2.41)). We tried to do that, but it seems unlikely that large values of these cut-offs will result in vanishing residues. Since they control the degrees of the poles, they will result in residues with higher derivatives of the Q-functions, which in turn will generate higher conserved charges in the residues, that themselves need to be canceled.

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
