# Peer review of "Orthogonality of Q-Functions up to Wrapping in Planar N = 4 Super Yang–Mills Theory"

_SciPost Physics_

## Round 1 · Referee Report · Anonymous (Referee 1) · 2025-9-26

Strengths

The authors of this paper propose orthogonality condition for the Q-functions in the sl(2) sector of planar N = 4 supersymmetric Yang-Mills theory which apply to all loop orders prior wrapping. These new SoV-type expressions are twist-independent and generalisable to different twists, although they do not account for degeneracies among states with equal spin. The proposed multiple integral is not yet a scalar product, because at each order it gives different normalisation factors when compared to the Gaudin norm.

The construction of scalar products is the most difficult part of the SoV approach to the computation of the correlation functions. With the complete solution still out of sight, any novel, even partial as is the case here, result is welcome and worth sharing.

Weaknesses

The paper's format is overly similar to a research logbook. Readers primarily interested in the final result may find this structure distracting.

Report

The construction of scalar products is the most difficult part of the SoV approach to the computation of the correlation functions. With the complete solution still out of sight, any novel, even partial as is the case here, result is welcome and worth sharing.

Requested changes

no

Recommendation

Publish (meets expectations and criteria for this Journal)

---

## Round 1 · Referee Report · Anonymous (Referee 2) · 2025-9-29

Disclosure of Generative AI use

The referee discloses that the following generative AI tools have been used in the preparation of this report:

Gemini 2.5 Flash, to help with the English language and smooth out the sentences (which in the end was not very useful since I had to come back and rework the output)

Strengths

  1. the paper makes progress in a field where progress has been very slow
  2. it cures some of the problems of the previous, three-year old attempt to solve the problem

Weaknesses

  1. the method provided doesn't work for states with degenerate spin that should be orthogonal
  2. a clear overall picture is still to emerge

Report

The paper details an attempt to find a Separation of Variables (SoV) representation for the sl(2) sector of the AdS/CFT integrable model, extending methods that have been proven successful for nearest-neighbor spin chains, as well as a three-years old previous work involving the first few orders in perturbation theory.
The authors' approach starts from the Baxter equation and involves dressing the Baxter polynomial with a non-polynomial factor raised to a power α whose relevant values are 0, 1/2, and 1. The paper uses a blend of analytical and "numerical" methods to search for SoV measures insuring orthogonality of Baxter functions for different states.
The main achievement is the successful derivation of such SoV measures order by order in perturbation theory until the wrapping order. The method works by increasing the spin chain's effective length by one with each increase in the perturbation order, and it cures some of problems of the previous attempt.
Despite the progress it allowed, the approach suffers of several drawbacks. The method doesn't work for states with degenerate spin that should be orthogonal. The role of the dressing factor and the parameter α remains unclear, as the various attempts and computations haven't fully elucidated their role.
The authors have made an effort towards a pedagogical presentation. However, since a clear idea how the SoV works for this model is still to emerge, the authors chose a presentation following their trials an errors and the logical flow is not always easy to follow.
Considering the path forward, the right way to think may come from considering, as the author suggest, all the sectors at a time and the way the SoV method relates with the ingredients of the Quantum Spectral Curve. Since the SoV for higher-rank nearest-neighbor chains is now relatively well understood, including some aspects of gl(N∣M), further effort could be invested in studying longer-range deformations of these higher-rank sectors.

The SoV method for non-compact, sl(2) based spin chains was worked out in the beginning of the years 2000 in a series of works by Derkachov, Korchemsky and Manashov. While these papers are cited, I think their importance is not properly acknowledged.

In conclusion, this is an interesting piece of work that makes incremental progress in a field where there was little progress recently. While many points remain to be worked out and a clear picture is still to be completed, the paper provides a useful basis for further efforts. Therefore, I recommend the paper for publication in SciPost.

Requested changes

The SoV method for non-compact, sl(2) based spin chains was worked out in the beginning of the years 2000 in a series of works by Derkachov, Korchemsky and Manashov. While these papers are cited, I think their importance should be better acknowledged.

Recommendation

Publish (easily meets expectations and criteria for this Journal; among top 50%)

---

## Round 2 · Referee Report · Anonymous (Referee 2) · 2025-11-26

Strengths

  1. the paper makes progress in a field where progress has been very slow

  2. it cures some of the problems of the previous, three-year old attempt to solve the problem

Weaknesses

  1. the method provided doesn't work for states with degenerate spin that should be orthogonal

  2. a clear overall picture is still to emerge

Report

The authors addressed my suggestions from the previous version of the report, therefore I recommend the paper for publication in SciPost

Requested changes

None

Recommendation

Publish (easily meets expectations and criteria for this Journal; among top 50%)

---

## Round 2 · Referee Report · Anonymous (Referee 1) · 2025-12-18

Report

In this paper, orthogonality relations are proposed for the sl(2) sector of planar N = 4 supersymmetric Yang-Mills theory. The authors construct universal measures and enlarged matrices of Q-functions that ensure distinct states vanish across all orders of perturbation theory, prior to considering wrapping corrections.
These new SoV-type expressions are twist-independent and generalisable to higher orders and different twists, although they currently do not account for degeneracies among states with equal spin.

However, the proposed multiple integral is not yet a scalar product, because at each order it gives different normalisation factors when compared to the Gaudin norm.
Another problem is that the proposed expression does not vanish when the two
Q-functions have the same spin but different other charges.
The authors discuss several possible ways to resolve these issues.

I think that the this results of this paper, although partial, might give insights
about how to develop the SoV approach to the N=4 SYM for finite coupling.
I recommend publication

Recommendation

Publish (easily meets expectations and criteria for this Journal; among top 50%)

---

## Round 2 · List of Changes

• As requested by one of the referees, we highlighted the importance of two papers already cited ([21],[22] in the new labeling). In the introduction we changed

"... the Separation of Variables (SoV) method, which was pioneered by Sklyanin [20] and has undergone rapid advancement in recent years ..." to "the Separation of Variables (SoV) method, which was pioneered by Sklyanin [20] and extensively developed in sl(2)-based models in [21, 22]. The formalism has undergone rapid advancement in recent years ..."

  • Added a new sentence to the Acknowledgements.
  • Minor grammar / spelling / reformulation of one or two sentences.
  • Added a few additional references.

---

## Editorial Decision

in_refereeing